# RECOVERING TOP-TWO ANSWERS AND CONFUSION PROBABILITY IN MULTI-CHOICE CROWDSOURCING

## ABSTRACT

Crowdsourcing has emerged as an effective platform to label a large volume of data in a cost- and time-efficient manner. Most previous works have focused on designing an efficient algorithm to recover only the ground-truth labels of the data. In this paper, we consider multi-choice crowdsourced labeling with the goal of recovering not only the ground truth but also the most confusing answer and the confusion probability. The most confusing answer provides useful information about the task by revealing the most plausible answer other than the ground truth and how plausible it is. To theoretically analyze such scenarios, we propose a model where there are top-two plausible answers for each task, distinguished from the rest of choices. Task difficulty is quantified by the confusion probability between the top two, and worker reliability is quantified by the probability of giving an answer among the top two. Under this model, we propose a two-stage inference algorithm to infer the top-two answers as well as the confusion probability. We show that our algorithm achieves the minimax optimal convergence rate. We conduct both synthetic and real-data experiments and demonstrate that our algorithm outperforms other recent algorithms. We also show the applicability of our algorithms in inferring the difficulty of tasks and training neural networks with the soft labels composed of the top-two most plausible classes.

## 1 INTRODUCTION

Crowdsourcing has been widely adopted to solve a large number of tasks in a time- and cost-efficient manner with the aid of human workers. In this paper, we consider 'multiple-choice' tasks where a worker is asked to provide a single answer among multiple choices. Some examples are as follows: 1) Using crowdsourcing platforms such as MTurk, we solve object counting or classification tasks on a large collection of images. Answers can be noisy either due to the difficulty of the scene or due to unreliable workers who provide random guesses. 2) Scores are collected from reviewers for papers submitted at a conference. For certain papers, scores can vary widely among reviewers, either due to the paper's inherent nature (clear pros and cons) or due to the reviewer's subjective interpretation of the scoring scale (Stelmakh et al., 2019; Liu et al., 2022).

In the above scenarios, responses provided by human workers may not be consistent among themselves not only due to the existence of unreliable workers but also due to the inherent difficulty of the tasks. In particular, for multiple-choice tasks, there could exist plausible answers other than the ground truth, which we call *confusing answers*.[1] For tasks with confusing answers, even reliable workers may provide wrong answers due to confusion. Thus, we need to decompose the two different causes of wrong answers: low reliability of workers and confusion due to task difficulty.

Most previous models for multi-choice crowdsourcing, however, fall short of modeling the confusion. For example, in the single-coin Dawid-Skene model (Dawid & Skene, 1979), which is the most widely studied crowdsourcing model in the literature, it is assumed that a worker is associated with a single skill parameter fixed across all tasks, which models the probability of giving a correct answer for every task. Under this model, any algorithm that infers the worker skill would count a confused labeling as the worker's error and lower its accuracy estimate for the worker, which results in a wrong estimate for their true skill level.

---

[1] This phenomenon is evident on public datasets: for 'Web' dataset (Zhou et al., 2012), which has five labels, the most dominating top-two answers take 80% of the overall answers and the ratio between the top two is 2.4:1.

To model the effect of confusion in multi-choice crowdsourcing problems, we propose a new model under which each task can have a confusing answer other than the ground truth, with a varying confusion probability across tasks. The task difficulty is quantified by the confusion probability, and the worker skill is modeled by the probability of giving an answer among the top two, to distinguish reliable workers from pure spammers who just provide random guesses among possible choices. We justify the proposed top-two model with public datasets. Under this new model, we aim to recover both the ground truth and the most confusing answer with the confusion probability, indicating how plausible the recovered ground truth is compared to the most confusing answer.

We provide an efficient two-stage inference algorithm to recover the top-two plausible answers and the confusion probability. The first stage of our algorithm uses the spectral method to get an initial estimate for top-two answers as well as the confusion probability, and the second stage uses this initial estimate to estimate the worker reliabilities and to refine the estimates for the top-two answers. Our algorithm achieves the minimax optimal convergence rate. We then perform experiments where we compare our method to recent crowdsourcing algorithms on both synthetic and real datasets, and show that our method outperforms other methods in recovering top-two answers. This result demonstrates that our model better explains the real-world datasets including errors from confusion. Our key contributions can be summarized as follows.

- *Top-two model:* We propose a new model for multi-choice crowdsourcing tasks where each task has top-two answers and the difficulty of the task is quantified by the confusion probability between the top-two. We justify the proposed model by analyzing six public datasets, and showing that the top-two structure explains well the real datasets.

- *Inference algorithm and its applicaitons:* We propose a two-stage algorithm that recovers the top-two answers and the confusion probability of each task at the minimax optimal convergence rate. We demonstrate the potential applications of our algorithm not only in crowdsourced labeling but also in quantifying task difficulty and training neural networks for classification with soft labels including the top-two information and the task difficulty.

**Related works** In crowdsourcing (Welinder et al., 2010; Liu & Wang, 2012; Demartini et al., 2012; Aydin et al., 2014; Demartini et al., 2012), one of the most widely studied models is the Dawid-Skene (D&S) model (Dawid & Skene, 1979). In this model, each worker is associated with a single confusion matrix fixed across all tasks, which models the probability of giving a label $b \in [K]$ for the true label $a \in [K]$ for $K$-ary classification task. In the single-coin D&S model, the model is further simplified such that each worker possesses a fixed skill level regardless of the true label or the task. Under the D&S model, various methods were proposed to estimate the confusion matrix or skill of each worker by spectral method (Zhang et al., 2014; Dalvi et al., 2013; Ghosh et al., 2011; Karger et al., 2013), belief propagation or iterative algorithms (Karger et al., 2014; 2011; Li & Yu, 2014; Liu et al., 2012; Ok et al., 2016), or rank-1 matrix completion (Ma et al., 2018; Ma & Olshevsky, 2020; Ibrahim et al., 2019). The estimated skill can be used to infer the ground-truth answer by approximating the maximum likelihood (ML)-type estimators (Gao & Zhou, 2013; Gao et al., 2016; Zhang et al., 2014; Karger et al., 2013; Li & Yu, 2014; Raykar et al., 2010; Smyth et al., 1994; Ipeirotis et al., 2010; Berend & Kontorovich, 2014). In contrast to the D&S models, our model allows the worker to have different probability of error caused by confusion. Thus, our algorithm needs to estimate not only the worker skill but also the task difficulty. Since the number of tasks is often much larger than the number of workers in practice, estimating the task difficulties is much more challenging than estimating worker skills. We provide a statistically-efficient algorithm to estimate the task difficulties and use this estimate to infer the top-two answers.

We also remark that there are some recent attempts to model task difficulties (Khetan & Oh, 2016; Shah et al., 2020; Krivosheev et al., 2020; Shah & Lee, 2018; Bachrach et al., 2012; Li et al., 2019; Tian & Zhu, 2015). However, these works are either restricted to binary tasks (Khetan & Oh, 2016; Shah et al., 2020; Shah & Lee, 2018) or focus on grouping confusable classes (Krivosheev et al., 2020; Li et al., 2019; Tian & Zhu, 2015). Our result, on the other hand, applies to any set of multi-choice tasks, where the choices of each task are not necessarily restricted to a fixed set of classes.

**Notation.** For a vector $\boldsymbol{x}$, $x_i$ represents the $i$-th component of $\boldsymbol{x}$. For a matrix $\boldsymbol{M}$, $M_{ij}$ refers to the $(i, j)$th entry of $\boldsymbol{M}$. For any vector $\boldsymbol{x}$, its $\ell_2$ and $\ell_\infty$-norm are denoted by $\|\boldsymbol{x}\|_2$ and $\|\boldsymbol{x}\|_\infty$, respectively. We follow the standard definitions of asymptotic notations, $\Theta(\cdot)$, $O(\cdot)$, $o(\cdot)$, and $\Omega(\cdot)$.

## 2 MODEL AND PROBLEM SETUP

We consider a crowdsourcing model to infer the top-two most plausible answers among $K$ choices for each task. There are $n$ workers and $m$ tasks. For each task $j \in [m] := \{1, \ldots, m\}$, we denote the correct answer by $g_j \in [K]$ and the next plausible, or the most confusing answer by $h_j \in [K]$. We call the pair $(g_j, h_j)$ the top-two answers for task $j \in [m]$. Let $\boldsymbol{p} \in [0, 1]^n$ and $\boldsymbol{q} \in (1/2, 1]^m$ be parameters modeling the reliability of workers and difficulty of tasks, respectively. For every pair of $(i, j)$, the $j$-th task is assigned to the $i$-th worker independently with probability $s$. We use a matrix $\boldsymbol{A} \in \mathbb{R}^{n \times m}$ to represent the responses of workers, where $A_{ij} = 0$ if the $j$-th task is not assigned to the $i$-th worker, and if it is assigned, and $A_{ij}$ is equal to the received label. The distribution of $A_{ij}$ is specified by the worker reliability $p_i$ and task difficulty $q_j$ as follows:

$$
A_{ij} = \begin{cases}
g_j, & \text{with prob. } s\left(p_i q_j + \frac{1-p_i}{K}\right), \\
h_j, & \text{with prob. } s\left(p_i(1 - q_j) + \frac{1-p_i}{K}\right), \\
\text{each } b \in [K] \backslash \{g_j, h_j\}, & \text{with prob. } s\left(\frac{1-p_i}{K}\right), \\
0, & \text{with prob. } 1 - s.
\end{cases} \tag{1}
$$

Here $p_i$ stands for the reliability of the $i$-th worker, in giving the answer from the most plausible top two $(g_j, h_j)$. If $p_i = 0$, the worker is considered a spammer who provides random answers among $K$ choices, and a larger value of $p_i$ indicates a higher worker reliability. The parameter $q_j$ represents the inherent difficulty of the task $j$ in distinguishing between the top two answers: for an easy task, $q_j$ is closer to 1, and for a hard task, $q_j$ is closer to 1/2. We call $q_j$ the confusion probability. Our goal is to recover top-two answers $(g_j, h_j)$ for all $j \in [m]$ with high probability with the minimum possible sampling probability $s$. We assume that the model parameters $(\boldsymbol{p}, \boldsymbol{q})$ are unknown.

We propose the top-two model to reflect common attributes of public crowdsourcing datasets, summarized in Appendix §A. The most important observation is that the top-two answers dominate the overall answers, and only the second-dominating answer has an incidence rate comparable to that of the ground truth. In other words, the standard deviation in the incidence rate of the second dominating answer has an overlap with that of the ground truth, but not the third-, or fourth-dominating answers. This indicates that assuming a unique 'confusing answer' is sufficient to model the confusion stemming from task difficulty. More details are available in Appendix §A.

**Binary conversion.** The $K$-ary task can be decomposed into $(K - 1)$-binary tasks (Karger et al., 2013): define $\boldsymbol{A}^{(k)}$ for $1 \le k < K$ such that the $(i, j)$-th entry $A_{ij}^{(k)}$ indicates whether the answer $A_{ij}$ is larger than $k$, i.e., $A_{ij}^{(k)} = -1$ if $1 \le A_{ij} \le k$; $A_{ij}^{(k)} = 1$ if $k < A_{ij} \le K$; and $A_{ij}^{(k)} = 0$ if $A_{ij} = 0$. We show that $\mathbb{E}[\boldsymbol{A}^{(k)}]$ is rank-1 and the singular value decomposition (SVD) of $\mathbb{E}[\boldsymbol{A}^{(k)}]$ can reveal the top-two answers $\{(g_j, h_j)\}_{j=1}^m$ and the confusion probability vector $\boldsymbol{q}$.

**Proposition 1.** *For every $1 \le k < K$, the binary-mapped matrix $\boldsymbol{A}^{(k)} \in \{-1, 0, 1\}^{n \times m}$ satisfies $\mathbb{E}[\boldsymbol{A}^{(k)}] - \frac{s(K-2k)}{K}\mathbb{1}_{n \times m} = 2s\boldsymbol{p}(\boldsymbol{r}^{(k)})^\top$, where $\boldsymbol{r}^{(k)} = [r_1^{(k)} \cdots r_m^{(k)}]^\top$ is defined as*

*Case I: $g_j > h_j$*          *Case II: $g_j < h_j$*

$$
r_j^{(k)} := \begin{cases}
\frac{k}{K} & \text{where } k < h_j; \\
\frac{k}{K} - (1 - q_j) & \text{where } h_j \le k < g_j; \\
\frac{k}{K} - 1 & \text{where } g_j \le k,
\end{cases}
\qquad
r_j^{(k)} := \begin{cases}
\frac{k}{K} & \text{where } k < g_j; \\
\frac{k}{K} - q_j & \text{where } g_j \le k < h_j; \\
\frac{k}{K} - 1 & \text{where } h_j \le k.
\end{cases}
$$

By defining $\Delta r_j^{(k)} := r_j^{(k)} - r_j^{(k-1)}$ for $k \in [K]$ with $r_j^{(0)} := 0$ and $r_j^{(K)} := 0$ for all $j$, we have

$$
\Delta r_j^{(k)} = \begin{cases}
\frac{1}{K} - q_j & \text{where } k = g_j, \\
\frac{1}{K} - (1 - q_j) & \text{where } k = h_j, \\
\frac{1}{K} & \text{otherwise.}
\end{cases} \tag{2}
$$

Note that $\Delta r_j^{(k)}$ has its minimum at $k = g_j$ and the second smallest value at $k = h_j$ for $q_j \in (1/2, 1]$. If one can specify $g_j$, the task difficulty $q_j$ can also be revealed from $\frac{1}{K} - \Delta r_j^{(g_j)}$. In the next section, we use this structure of $\boldsymbol{r}^{(k)}$ for $k \in [K]$ to infer the top two answers and the confusion probability.[2]

---

[2] We assume that $\eta\sqrt{n} \le \|\boldsymbol{p}\|_2 \le \sqrt{n}$ for some $\eta > 0$, i.e., there are only $o(n)$ spammers ($p_i = 0$), and $\|\boldsymbol{r}^{(k)}\|_2 = \Theta(\sqrt{m})$ for every $k \in [K]$, which can be easily satisfied except exceptional cases from equation 2.

---

**Algorithm 1** Spectral Method for Initial Estimation (TopTwo1 Algorithm)

---

1: **Input:** data matrix $\boldsymbol{A}^1 \in \{0, 1, \dots, K\}^{n \times m}$ and parameter $\eta > 0$ where $\eta \sqrt{n} \leq \|\boldsymbol{p}\|_2 \leq \sqrt{n}$.

2: Randomly split (with equal probabilities) and convert $\boldsymbol{A}^1$ into binary matrices $\boldsymbol{X}^{(k)} \in \{-1, 0, 1\}^{n \times m}$ and $\boldsymbol{Y}^{(k)} \in \{-1, 0, 1\}^{n \times m}$ for $1 \leq k < K$ as described in Sec. 3.1.

3: Let $\boldsymbol{u}^{(k)}$ be the leading normalized left singular vector of $\boldsymbol{X}^{(k)}$. Trim the abnormally large components of $\boldsymbol{u}^{(k)}$ by letting it be zero if $u_i^{(k)} > \frac{2}{\eta \sqrt{n}}$ and denote the resulting vector as $\tilde{\boldsymbol{u}}^{(k)}$.

4: Calculate the estimate of $\|\boldsymbol{p}\|\boldsymbol{r}^{(k)}$ by $\boldsymbol{v}^{(k)} := \frac{1}{s'}(\boldsymbol{Y}^{(k)})^\top \tilde{\boldsymbol{u}}^{(k)}$. Assume $\boldsymbol{v}^{(0)} := \boldsymbol{0}$ and $\boldsymbol{v}^{(K)} := \boldsymbol{0}$.

5: For $k \in [K]$, calculate $\Delta v_j^{(k)} := v_j^{(k)} - v_j^{(k-1)}$. Estimate the top-two answers for $j \in [m]$ by

$$\hat{g}_j := \arg\min_{k \in [K]} \Delta v_j^{(k)}; \quad \hat{h}_j := \arg\min_{k \neq \hat{g}_j, k \in [K]} \Delta v_j^{(k)}. \tag{3}$$

6: Estimate $\|\boldsymbol{p}\|_2$ by defining $l_j := \frac{K}{K-2} \sum_{k \neq \hat{g}_j, k \neq \hat{h}_j} \Delta v_j^{(k)}$ and $l := \frac{1}{m} \sum_{j=1}^m l_j$.

7: Estimate $q_j$ for $j \in [m]$ by defining

$$\hat{q}_j := 1/K - \Delta v_j^{(\hat{g}_j)}/l. \tag{4}$$

8: **Output:** estimated top-two answers $\{(\hat{g}_j, \hat{h}_j)\}_{j=1}^m$ and confusion probability vector $\hat{\boldsymbol{q}}$.

---

## 3 PROPOSED ALGORITHM

Our algorithm consists of two stages. In Stage 1, we compute an initial estimate on top-two answers and the confusion probability $\boldsymbol{q}$. In Stage 2, we estimate the worker reliability vector $\boldsymbol{p}$ by using the result of the first stage, and use the estimated $\boldsymbol{p}$ and $\boldsymbol{q}$ to refine our estimates for the top two answers. Assume that we randomly split the original response matrix $\boldsymbol{A}$ into $\boldsymbol{A}^1$ and $\boldsymbol{A}^2$ with probability $s_1$ and $1 - s_1$, respectively, and use only $\boldsymbol{A}^1$ for stage 1 and $(\boldsymbol{A}^1, \boldsymbol{A}^2)$ for stage 2.

### 3.1 STAGE 1: INITIAL ESTIMATES USING SVD

The first stage begins with randomly splitting $\boldsymbol{A}^1$ again into two independent matrices $\boldsymbol{B}$ and $\boldsymbol{C}$ with equal probabilities. We then convert $\boldsymbol{B}$ and $\boldsymbol{C}$ into $(K-1)$-binary matrices $\boldsymbol{B}^{(k)}$ and $\boldsymbol{C}^{(k)}$ as explained in Sec. 2. Define $\boldsymbol{X}^{(k)}$ and $\boldsymbol{Y}^{(k)}$ as $\boldsymbol{X}^{(k)} := \boldsymbol{B}^{(k)} - \frac{s'(K-2k)}{K} \mathbb{1}_{n \times m}$ and $\boldsymbol{Y}^{(k)} := \boldsymbol{C}^{(k)} - \frac{s'(K-2k)}{K} \mathbb{1}_{n \times m}$ for $s' = s \cdot s_1/2$. We have $\mathbb{E}[\boldsymbol{X}^{(k)}] = \mathbb{E}[\boldsymbol{Y}^{(k)}] = s' \boldsymbol{p} (\boldsymbol{r}^{(k)})^\top$ from Prop. 1.

We use $\boldsymbol{X}^{(k)}$ and $\boldsymbol{Y}^{(k)}$ to estimate $\boldsymbol{p}^* := \boldsymbol{p}/\|\boldsymbol{p}\|_2$ and $\|\boldsymbol{p}\|_2 \boldsymbol{r}^{(k)}$, respectively. The estimators are denoted by $\boldsymbol{u}^{(k)}$ and $\boldsymbol{v}^{(k)}$, respectively. We define $\boldsymbol{u}^{(k)}$ as the left singular vector of $\boldsymbol{X}^{(k)}$ with the largest singular value. Sign ambiguity of the singular vector is resolved by defining $\boldsymbol{u}^{(k)}$ as the one between $\{\boldsymbol{u}^{(k)}, -\boldsymbol{u}^{(k)}\}$ in which at least half of the entries are positive. After trimming abnormally large components of $\boldsymbol{u}^{(k)}$ and defining the trimmed vector as $\tilde{\boldsymbol{u}}^{(k)}$, we calculate $\boldsymbol{v}^{(k)} := \frac{1}{s'}(\boldsymbol{Y}^{(k)})^\top \tilde{\boldsymbol{u}}^{(k)}$, which is an estimate for $\|\boldsymbol{p}\|_2 \boldsymbol{r}^{(k)}$. By using $\boldsymbol{v}^{(k)}$ for $1 \leq k < K$, we get estimates for top-two answers $(\hat{g}_j, \hat{h}_j)$ based on the observation in equation 2. Lastly, we estimate $\|\boldsymbol{p}\|_2$ and use $\boldsymbol{v}^{(k)}/\|\boldsymbol{p}\|_2 \approx \boldsymbol{r}^{(k)}$ to estimate the confusion probability vector $\boldsymbol{q}$. See Algorithm 1 for details.

### 3.2 STAGE 2: PLUG-IN MAXIMUM LIKELIHOOD ESTIMATOR (MLE)

The second stage uses the result of Stage 1 to estimate the worker reliability vector $\boldsymbol{p}$. We first propose an estimate for the worker reliability vector $\boldsymbol{p}$ by using the estimated top-two answers $\{(g_j, h_j)\}_{j=1}^m$ from Algorithm 1. We randomly split the original response matrix $\boldsymbol{A}$ into $\boldsymbol{A}^1$ and $\boldsymbol{A}^2$ with probability $s_1$ and $1 - s_1$, respectively, and use $\boldsymbol{A}^1$ only for Algorithm 1 and $\boldsymbol{A}^2$ only for calculating the estimator $\hat{\boldsymbol{p}}$. Our estimate for the worker reliability $p_i$ is defined as

$$\hat{p}_i = \frac{K}{(K-2)} \left( \frac{1}{s(1-s_1)} \left( \frac{1}{m} \sum_{j=1}^m \mathbb{1}(A_{ij}^2 = \hat{g}_j \text{ or } \hat{h}_j) \right) - \frac{2}{K} \right). \tag{5}$$

---

**Algorithm 2** Plug-in MLE (TopTwo2 Algorithm)

1: **Input:** data matrix $\boldsymbol{A} \in \{0, 1, \ldots, K\}^{n \times m}$ and the sample splitting rate $s_1 > 0$.
2: Randomly split $\boldsymbol{A}$ into $\boldsymbol{A}^1$ and $\boldsymbol{A}^2$ by defining $\boldsymbol{A}^1 := \boldsymbol{A} \circ \boldsymbol{S}$ and $\boldsymbol{A}^2 = \boldsymbol{A} \circ (\mathbb{1}_{n \times m} - \boldsymbol{S})$ where $\boldsymbol{S}$ is an $n \times m$ matrix whose entries are i.i.d. with Bern$(s_1)$ and $\circ$ is an entrywise product.
3: Apply Algorithm 1 to $\boldsymbol{A}^1$ to yield estimates for top-two answers $\{(\hat{g}_j, \hat{h}_j)\}_{j=1}^m$ and confusion probability vector $\hat{\boldsymbol{q}}$.
4: By using $\{(\hat{g}_j, \hat{h}_j)\}_{j=1}^m$ and $\boldsymbol{A}^2$, calculate the estimate $\hat{\boldsymbol{p}}$ as in equation 5.
5: By using the whole $\boldsymbol{A}$ and $(\hat{\boldsymbol{p}}, \hat{\boldsymbol{q}})$, find the plug-in MLE estimates $(\hat{g}_j^{\mathsf{MLE}}, \hat{h}_j^{\mathsf{MLE}})$ by

$$\arg\max_{a,b \in [K]^2, a \neq b} \sum_{i=1}^n \log\left(\frac{K\hat{p}_i \hat{q}_j}{1 - \hat{p}_i} + 1\right) \mathbb{1}(A_{ij} = a) + \log\left(\frac{K\hat{p}_i(1 - \hat{q}_j)}{1 - \hat{p}_i} + 1\right) \mathbb{1}(A_{ij} = b). \quad (6)$$

6: **Output:** estimated top-two answers $\{(\hat{g}_j^{\mathsf{MLE}}, \hat{h}_j^{\mathsf{MLE}})\}_{j=1}^m$.

---

Our plug-in MLE uses the estimated $(\hat{\boldsymbol{p}}, \hat{\boldsymbol{q}})$ in the place of $(\boldsymbol{p}, \boldsymbol{q})$ at the oracle MLE, which finds $(\hat{g}_j, \hat{h}_j) \in [K]^2 \backslash \{(1,1), (1,2), \ldots, (K,K)\}$ such that $(\hat{g}_j, \hat{h}_j) := \arg\max_{(a,b) \in [K]^2, a \neq b} \sum_{i=1}^n \log \mathbb{P}(A_{ij} | \boldsymbol{p}, q_j, (a,b))$ as in equation 6. Details can be found in Alg.2.

The time complexity of Alg. 2 is $O(m^2 \log m + nmK^2)$, since the SVD in Alg. 1 can be computed via power iterations within $O(m^2 \log m)$ steps (Boutsidis et al., 2015), and the step for finding the pair of answers maximizing equation 6 requires $O(nmK^2)$ steps.

## 4 PERFORMANCE ANALYSIS

To state our main theoretical results, we first need to introduce some notation and assumptions. Let $\mu_{(a,b),k}^{(i,j)}$ denote the probability that a worker $i \in [n]$ gives label $k \in [K]$ for the assigned task $j \in [m]$ of which the top-two answers are $(g_j, h_j) = (a, b)$. Note that $\mu_{(a,b),k}^{(i,j)}$ can be written in terms of $(p_i, q_j)$ from the distribution in equation 1 for every $a, b, k \in [K]^3$. Let $\boldsymbol{\mu}_{(a,b)}^{(i,j)} = [\mu_{(a,b),1}^{(i,j)} \quad \mu_{(a,b),2}^{(i,j)} \quad \cdots \quad \mu_{(a,b),K}^{(i,j)}]^\top$. We introduce a quantity that measures the average ability of workers in distinguishing the ground-truth pair of top-two answers $(g_j, h_j)$ from any other pair $(a, b) \in [K]^2 / \{(g_j, h_j)\}$ for the task $j \in [m]$. We define

$$\overline{D}^{(j)} := \min_{(g_j, h_j) \neq (a,b)} \frac{1}{n} \sum_{i=1}^n \mathbb{D}_{\mathsf{KL}}\left(\boldsymbol{\mu}_{(g_j, h_j)}^{(i,j)}, \boldsymbol{\mu}_{(a,b)}^{(i,j)}\right); \quad \overline{D} := \min_{j \in [m]} \overline{D}^{(j)}, \quad (7)$$

where $\mathbb{D}_{\mathsf{KL}}(P, Q) := \sum_i P(i) \log(P(i)/Q(i))$ is the KL-divergence between $P$ and $Q$. Note that $\overline{D}^{(j)}$ is strictly positive if there exist at least one worker $i$ with $p_i > 0$ and $q_j \in (1/2, 1)$ for the distribution in equation 1, so that $(g_j, h_j)$ can be distinguished from any other $(a, b) \in [K]^2 / \{(g_j, h_j)\}$ statistically. We define $\overline{D}$ as the minimum of $\overline{D}^{(j)}$ over $j \in [m]$, indicating the average ability of workers in distinguishing $(g_j, h_j)$ from any other $(a, b)$ for the most difficult task in the set of tasks.

We split the performance analysis of our algorithm into two parts. First, Theorem 1 states the performance guarantees for Alg. 1.

**Theorem 1** (Performance Guarantees for Algorithm 1). *For any $\epsilon, \delta_1 > 0$, if the sampling probability $s \cdot s_1 = \Omega\left(\frac{1}{\delta_1^2 \|\boldsymbol{p}\|_2^2} \log \frac{K}{\epsilon}\right)$, Algorithm 1 guarantees the recovery of the ordered top-two answers $(g_j, h_j)$ with probability at least $1 - \epsilon$ for any $j \in [m]$ with $q_j \in (1/2, 1)$, i.e.,*

$$\mathbb{P}\left((\hat{g}_j, \hat{h}_j) = (g_j, h_j)\right) \geq 1 - \epsilon \quad \text{for all } j \in [m] \text{ with } q_j \in (1/2, 1), \quad (8)$$

*and the recovery of the confusion probability $q_j$ with*

$$\mathbb{P}\left(|\hat{q}_j - q_j| < \delta_1\right) \geq 1 - \epsilon \quad \text{for all } j \in [m], \quad (9)$$

*for every sufficiently large number $m$ of tasks and the number of workers $n = O(m/\log m)$.*

By using Theorem 1, we can also find the sufficient conditions to guarantee the recovery of paired top-two answers for *all* tasks and $q$ with an arbitrarily small $\ell_\infty$-norm error.

**Corollary 1.** *For any $\epsilon, \delta_1 > 0$, if the sampling probability $s \cdot s_1 = \Omega\left(\frac{1}{\delta_1^2 \|p\|_2^2} \log \frac{mK}{\epsilon}\right)$, Algorithm 1 guarantees the recovery of $\{(g_j, h_j)\}_{j=1}^m$ and $q$ with probability at least $1 - \epsilon$ as $m \to \infty$ such that*

$$\mathbb{P}\left((\hat{g}_j, \hat{h}_j) = (g_j, h_j), \forall j \in [m]\right) \geq 1 - \epsilon \quad and \quad \mathbb{P}\left(\|q - \hat{q}\|_\infty < \delta_1\right) \geq 1 - \epsilon. \quad (10)$$

Proofs of Theorem 1 and Corollary 1 are available in Appendix §G.

We next analyze the performance of Alg. 2, which uses Alg. 1 as the first stage. Before providing the main theorem for Alg. 2, we state a lemma charactering a sufficient condition for estimating the worker reliability vector $p$ from equation 5 with an arbitrarily small $\ell_\infty$-norm error.

**Lemma 1.** *Conditioned on $(\hat{g}_j, \hat{h}_j) = (g_j, h_j)$ for all $j \in [m]$, if $s(1 - s_1) = \Omega\left(\frac{1}{\delta_2^2 m} \log \frac{n}{\epsilon}\right)$, the estimator $\hat{p}_i$ defined in equation 5 of Alg. 2 guarantees $\mathbb{P}\left(\|p - \hat{p}\|_\infty < \delta_2\right) \geq 1 - \epsilon$ for any $\epsilon > 0$.*

Combining Corollary 1 and Lemma 1, we can have the estimators $(\hat{p}, \hat{q})$ for the worker reliability vector $p$ and the confusion probability vector $q$ with $\ell_\infty$-norm error bounded by any arbitrarily small $\delta > 0$ with probaiblity at least $1 - 2\epsilon$ if

$$s = s \cdot s_1 + s(1 - s_1) = \Omega\left(\frac{\log(mK/\epsilon)}{\delta^2 \|p\|_2^2} + \frac{\log(n/\epsilon)}{\delta^2 m}\right) = \Omega\left(\frac{\log(mK/\epsilon)}{\delta^2 \|p\|_2^2}\right) \quad (11)$$

where the last equality is from the assumption that $\|p\|_2 = \Theta(\sqrt{n})$ and $n = O(m/\log m)$. In this regime, the sample complexity for estimating the task difficulty $q$ is larger than that for estimating worker reliability $p$. To make sure that the sampling probability $s < 1$, we need $n = \Omega(\log m)$.

Our second theorem, Theorem 2, characterizes the sufficient condition on the sampling probability $s$ to guarantee the recovery of the pair of top-two answers for all tasks by equation 6 of Alg. 2, when a sufficiently accurate estimation of $(p, q)$ is given.

**Theorem 2.** *Assume that there is a positive scalar $\rho$ such that $\mu_{(g_j, h_j), c}^{(i,j)} \geq \rho$ for all $(i, j, g_j, h_j, c) \in [n] \times [m] \times [K]^3$. For any $\epsilon > 0$, if $(\hat{p}, \hat{q})$ are given with*

$$\max\{\|p - \hat{p}\|_\infty, \|q - \hat{q}\|_\infty\} \leq \delta := \min\left\{\frac{\rho}{4}, \frac{\rho\overline{D}}{4(6 + \overline{D})}\right\}, \quad (12)$$

*and the sampling probability $s = \Omega\left(\frac{\log(1/\rho) \log(mK^2/\epsilon) + \overline{D} \log(m/\epsilon)}{n\overline{D}}\right)$, then for any $\epsilon > 0$ the estimates of $\{(g_j, h_j)\}_{j=1}^m$ from equation 6 of Algorithm 2 guarantees*

$$\mathbb{P}\left((\hat{g}_j, \hat{h}_j) = (g_j, h_j), \forall j \in [m]\right) \geq 1 - \epsilon. \quad (13)$$

Proofs of Lemma 1 and Theorem 2 are available in Appendix §H. The assumption in Theorem 2 that $\mu_{(g_j, h_j), c}^{(i,j)} \geq \rho$ for some $\rho > 0$ holds if $p_i < 1$ for all $i \in [n]$, i.e., there is no perfectly reliable worker. This assumption can be easily satisfied by adding an arbitrary small random noise to the worker answers as well. By combining the statements in Corollary 1, Lemma 1, and Theorem 2 with $\delta_1 = \delta_2 = \delta$ for $\delta$ defined in equation 12, we get the overall performance guarantee for Alg. 2.

**Corollary 2** (Performance Guarantees for Alg. 2). *Alg. 2 guarantees the recovery of top-two answers for all tasks with $\mathbb{P}\left((\hat{g}_j, \hat{h}_j) = (g_j, h_j), \forall j \in [m]\right) \geq 1 - \epsilon$ for any $\epsilon > 0$ if $s$ satisfies*

$$s = \Omega\left(\frac{\log(mK/\epsilon)}{\delta^2 \|p\|_2^2} + \frac{\log(1/\rho) \log(mK^2/\epsilon) + \overline{D} \log(m/\epsilon)}{n\overline{D}}\right) = \Omega\left(\frac{\log(m/\epsilon)}{\delta^2 \|p\|_2^2} + \frac{\log(m/\epsilon)}{n\overline{D}}\right). \quad (14)$$

In equation 14, the first term is for guaranteeing accurate estimates of $p$ and $q$ with $\ell_\infty$-norm error bounded by $\delta$ and the second term is for guaranteeing the recovery of the top-two answers from MLE with high probability. Since $\|p\|_2^2 = \Theta(n)$, the two terms effectively have the same order but with different constant scaling, depending on model-specific parameters $(p, q)$.

Lastly, we show the optimality of convergence rates of Alg. 1 and Alg. 2 with respect to two types of minimax errors, respectively. The proof of Theorem 3 is available in Appendix §I.

**Theorem 3.** *(a) Let $\mathcal{F}_{\overline{p}}$ be a set of $\boldsymbol{p} \in [0,1]^n$ such that the collective quality of workers, measured by $\|\boldsymbol{p}\|_2$, is parameterized by $\overline{p}$ as $\mathcal{F}_{\overline{p}} := \{\boldsymbol{p} : \frac{1}{n}\|\boldsymbol{p}\|_2^2 = \overline{p}\}$. Assume that $\overline{p} \leq 2/3$. If the average number $ns$ of samples (queries) per task is less than $(1/2\overline{p})\log(1/\epsilon)$, then*

$$\min_{\hat{\boldsymbol{g}}} \max_{\boldsymbol{p} \in \mathcal{F}_{\overline{p}}, \; \boldsymbol{g} \in [K]^m} \frac{1}{m} \sum_{j \in [m]} \mathbb{P}(\hat{g}_j \neq g_j) \geq \epsilon. \tag{15}$$

*(b) There is a universal constant $c > 0$ such that for any $\boldsymbol{p} \in [0,1]^n$, $\boldsymbol{q} \in (1/2, 1]^m$, if the sampling probability $s < \Omega\left(1/(n\overline{D})\right)$, then*

$$\min_{(\hat{\boldsymbol{g}}, \hat{\boldsymbol{h}})} \max_{\substack{(\boldsymbol{g}, \boldsymbol{h}) \in [K]^m \times [K]^m \\ g_j \neq h_j, \forall j [m]}} \frac{1}{m} \sum_{j \in [m]} \mathbb{P}((\hat{g}_j, \hat{h}_j) \neq (g_j, h_j)) \geq c. \tag{16}$$

From part (a) of Theorem 3, it is necessary to have $s > \Omega\left((1/\|\boldsymbol{p}\|_2^2)\log(1/\epsilon)\right)$ to make the minimax error in equation 15 less than $\epsilon$. Since Theorem 1 shows that Alg. 1 recovers $(\hat{g}_j, \hat{h}_j)$ with probability at least $1 - \epsilon$ if $s > \Omega\left((1/\|\boldsymbol{p}\|_2^2)\log(1/\epsilon)\right)$ when $s_1 = 1$, we can conclude that Alg. 1 achieves the minimax optimal rate for a fixed collective intelligence of workers, measured by $\|\boldsymbol{p}\|_2$. From part (b) of Theorem 3, for any $(\boldsymbol{p}, \boldsymbol{q})$, unless we have $s > \Omega(1/(n\overline{D}))$ there always exists a constant fraction of tasks for which the recovered top-two answers are incorrect. This bound matches with our sufficient condition on $s$ from Alg. 2 in equation 14 upto logarithmic factors, as long as $\delta^2\|\boldsymbol{p}\|_2 \gtrsim n\overline{D}$, showing the minimax optimality of our Alg. 2 for any $(\boldsymbol{p}, \boldsymbol{q})$. More discussions on the theoretical results are available at Appendix §E.

## 5 EXPERIMENTS

We evaluate the proposed algorithm under diverse scenarios of synthetic datasets in Sec. 5.1, and for two applications–in identifying difficult tasks in real datasets in Sec. 5.2 and in training neural network models with soft labels defined from the top-two plausible labels in Sec. 5.3.

### 5.1 EXPERIMENTS ON SYNTHETIC DATASET

We compare the empirical performance of Alg. 1 and Alg. 2 (referred as TopTwo1 and TopTwo2) with other baselines: majority voting(MV), OTP-D&S and MV-D&S (Zhang et al., 2014), PGD (Ma et al., 2018), M-MSR (Ma & Olshevsky, 2020) and oracle-MLE, whose details can be found in Appx. §C. We choose these baselines since they have the strongest established guarantees in the literature and they are all MLE-based approaches, from which the top-two answers can be inferred. Obviously, oracle-MLE, which uses the ground-truth model parameters, provides the best possible performance. We devise four scenarios described in Table 1 to verify the robustness of our model for various $(\boldsymbol{p}, \boldsymbol{q})$ ranges, at $(n, m) = (50, 500)$ with $s \in (0, 0.2]$. The number of choices for each task is fixed as 5. Fig. 1 reports the empirical error probability $\frac{1}{m}\sum_{j=1}^{m}\mathbb{P}((\hat{g}_j, \hat{h}_j) \neq (g_j, h_j))$ averaged over 30 runs, with 95% confidence intervals (shaded region). Four columns correspond to the four scenarios, resp. The prediction errors for $g_j$ and $h_j$ are plotted in Fig. 6 of Appx. §D.1.

We can observe that for all the considered scenarios TopTwo2 achieves the best performance, near the oracle MLE, in recovering $(g_j, h_j)$. Depending on the scenarios, the reason TopTwo2 outperforms can be explained differently. For the Easy scenario, since $q_j$ is close to 1, it is easy to distinguish $g_j$ from $h_j$ but hard to distinguish $h_j$ from other labels. Our algorithm achieves the best

Table 1: Parameters for synthetic data experiments under diverse scenarios.

| | Easy | Hard | Few-smart | High-variance |
|---|---|---|---|---|
| Worker | $p_i \in [0,1]$ | $p_i \in [0,1]$ | 90% $p_i \in [0,0.1]$ 
 10% $p_i \in [0.9,1]$ | $p_i \in [0,1]$ |
| Task | $q_j \in [0.9,1]$ | $q_j \in (0.5,0.6)$ | $q_j \in (0.5,1]$ | 50% $q_j \in (0.5,0.6)$ 
 50% $q_j \in [0.9,1.0]$ |

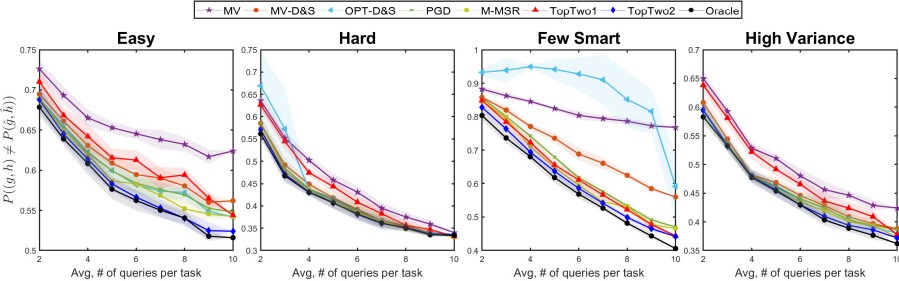

Figure 1: Prediction error for $(g, h)$ for four scenarios as the avg. number of queries per task changes. Our TopTwo2 alg. achieves the best performance, near the oracle MLE for all the scenarios.

performance in estimating $h_j$ by a large margin (Fig. 6). For the Hard scenario, it is hard to distinguish $g_j$ and $h_j$, but our algorithm, which uses an accurate $\hat{q}_j$, can better distinguish $g_j$ and $h_j$. For Few-smart, our algorithm achieves the biggest gain compared to other methods, since our algorithm can effectively distinguish few smart workers from spammers. High-variance shows the effect of having diverse $q_j$ in a dataset. We remark that our algorithm achieves the best performance, near that of the oracle-MLE, for all the scenarios, while the next performer keeps changing depending on scenarios. For example, the OPT D&S is the second best performer in the Easy scenario, while it is the worst performer in the Few-smart scenario. We also show the robustness of our algorithm against changes in model parameters in Appendix §D.

## 5.2 EXPERIMENTS ON REAL-WORLD DATASET: INFERRING TASK DIFFICULTIES

We next provide experimental results using real-world data collected from MTurk and show that our algorithm can be used for inferring task difficulties. We devised a color comparison task where we asked the crowd to choose a color, among six given choices, that looks the most similar to a reference color of each task. See Fig. 4 in Appx. §A.1 for example tasks. After randomly generating a reference color and the six choices, we identified the ground truth and the most confusing answer for each task by measuring the distance between colors using the CIEDE2000 color difference formula (Sharma et al., 2005). If the distance from the reference color to the ground truth is much shorter than that to the most confusing answer, then the task was considered easy. We designed 1000 tasks and distributed it to 200 workers, collecting 19.5 responses on each task. After collecting the data, we subsampled it to simulate how the prediction error decreases as the number of responses per task increases. Fig. 2a shows the performances in detecting $(g_j, h_j)$, $g_j$ and $h_j$, averaged over 10 times of random sampling, with 95% confidence interval (shaded region). TopTwo2 algorithm achieved the best performance in detecting $(g_j, h_j)$, $g_j$ and $h_j$ in all ranges. We further examined the correlation between the task difficulty - quantified by the distance gap between the ground truth and the most

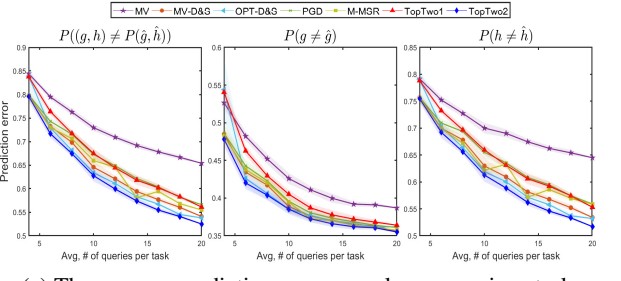

(a) The average prediction error on color comparison tasks

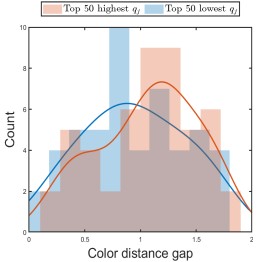

(b) Histogram of dist. gap

Figure 2: (a) Prediction error for $(g_j, h_j)$, $g_j$ and $h_j$ (from left to right) for color comparison tasks using real data collected from MTurk. Our TopTwo2 algorithm achieves the best performance. (b) Histogram of color distance gap for the task groups with the highest $q_j$ (easiest tasks) and lowest $q_j$ (most difficult tasks). The difficult task group (blue) tends to have a smaller color distance gap.

confusing answer from the reference color - and the estimated confusion probability $q_j$ across tasks. We selected top 50 most difficult/easiest tasks according to the estimated confusion probability $q_j$ and plotted the histograms of the distance gap for the two groups in Fig 2b. We can see that the difficult group (blue, having lowest $q_j$) tends to have a smaller distance gap than those of the easy task group (red). This result shows that our algorithm can identify difficult tasks in real datasets.

## 5.3 TRAINING NEURAL NETWORKS WITH SOFT LABELS HAVING TOP-TWO INFORMATION

An appealing example where we can use the knowledge of the second best answer is in training deep neural networks for classification tasks. Traditionally, a hard label (one ground-truth label per image) has been used to train a classifier. In recent works, it has been shown that using a soft label (full label distribution that reflect human perceptual uncertainty) is sometimes beneficial in obtaining a model with better generalization capability (Peterson et al., 2019). However, obtaining an accurate full label distribution requires much higher sample complexity than recovering only the ground-truth. For example, Peterson et al. (2019) provided a CIFAR10H dataset with full human label distributions for 10000 instances of CIFAR10 test examples by collecting on average 50 judgements per image, which is about 5-10 times larger than those of usual datasets (Table 4 in Appendix A.1).

Our top-two model, on the other hand, can effectively reduce the required sample complexity, while still guaranteeing the advantages in training models with soft labels. To demonstrate this idea, we trained two deep neural networks, VGG-19 and ResNet18, with the soft-label vectors having the top-two structure (top2) for CIFAR10H dataset[3]. We then compared the training/test results with those of the hard label (hard) and full label distribution (full). Experimental details are in Appendix §B. Compared to the original training with hard labels, training with top-two soft labels achieved 1.56% and 4.09% higher test accuracy in VGG-19 and ResNet18, respectively (averaged in three runs, 150 epochs) as shown in Table 2, which is even higher than that of the full label distribution in VGG-19. This result shows that training with the top-two soft labels results in better generalization (test accuracy) than training with hard labels, since the top-two soft label includes simple yet helpful side information, the most confusable class and the confusion probability.

Table 2: Comparison of performances for CIFAR10H dataset with hard/soft label training

| Network | Train accuracy | Training loss | Test accuracy | Test loss |
|---|---|---|---|---|
| VGG-19 (hard) | **97.46±0.59%** | **0.081±0.012** | 77.64±1.54% | 1.057±0.118 |
| VGG-19 (top2) | 97.00±0.51% | 0.231±0.014 | **79.20±1.04%** | 0.754±0.050 |
| VGG-19 (full) | 96.69±0.48% | 0.282±0.010 | 78.66±0.97% | **0.740±0.030** |
| ResNet18 (hard) | 98.47±0.320% | **0.046±0.009** | 76.49%±1.80% | 1.275±0.157 |
| ResNet18 (top2) | 98.67±0.491% | 0.168±0.024 | 80.58%±2.36% | 0.640±0.093 |
| ResNet18 (full) | **99.19±0.125%** | 0.189±0.023 | **80.93%±2.66%** | **0.611±0.102** |

## 6 DISCUSSION

We proposed a new model for multiple-choice crowdsourcing, with top-two confusable answers and varying confusion probability over tasks. We provided an algorithm to infer the top-two answers and the confusion probability. This work can benefit several query-based data acquisition systems such as MTurk or review systems by providing additional information about the task such as the most plausible answer other than the ground truth and how plausible it is, which can be used to quantify the accuracy of the ground truth or to classify the tasks based on difficulty. The topic of confusion is getting increasing attention in the machine learning community for designing reliable classifiers (Jin et al., 2017; Luque et al., 2019; Chang et al., 2017). We also demonstrated possible applications of our algorithm in designing soft labels for better generalization of neural networks.

---

[3]As in (Peterson et al., 2019), we used the original 10000 test examples of CIFAR10 for training and 50000 training examples for testing. Thus, the final accuracy is lower than usual. Since CIFAR10H is collected from selected 'reliable' workers who answered a set of test examples with an accuracy higher than 75%, we directly used the top-two dominating answers and the fraction between the two in designing the soft label vector (top2).

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

# A VERIFICATION FOR THE PROPOSED TOP-TWO MODEL

We proposed the top-two model to reflect the key attributes of seven datasets including Adult2, Dog, Web, Flag, Food, Plot, and Color, of which the details are summarized in Appendix A.1.

Table 3 shows empirical distributions of the mean incidence of responses for the top-three dominating answers, sorted by the dominance proportions, for the six public datasets and the Color dataset that we collected, with the standard deviation over the tasks in the dataset. In Fig. 3, we also plot empirical distributions of the mean incidence of responses sorted by the dominant proportion with error bars indicating the standard deviation. The $i$-th data point represents the average incidence of the $i$-th highest response in each task. For example, in Adult2 dataset, the most dominating answer takes 0.8 portion of the total answers, and the next dominating answer takes 0.14 portion of the total answers on average.

Table 3: Proportions of top-three dominating answers in public datasets

| Dataset | Ground truth | 2nd dominating answer | 3rd dominating answer |
|---------|--------------|-----------------------|-----------------------|
| Adult2 | 0.80±0.19 | 0.14±0.13 | 0.04±0.07 |
| Dog | 0.76±0.15 | 0.22±0.14 | 0.01±0.04 |
| Web | 0.59±0.20 | 0.25±0.12 | 0.12±0.09 |
| Flag | 0.90±0.16 | 0.09±0.13 | 0.01±0.03 |
| Food | 0.80±0.18 | 0.17±0.15 | 0.02±0.05 |
| Plot | 0.62±0.21 | 0.30±0.16 | 0.06±0.07 |
| Color | 0.43±0.1 | 0.23±0.06 | 0.15±0.05 |

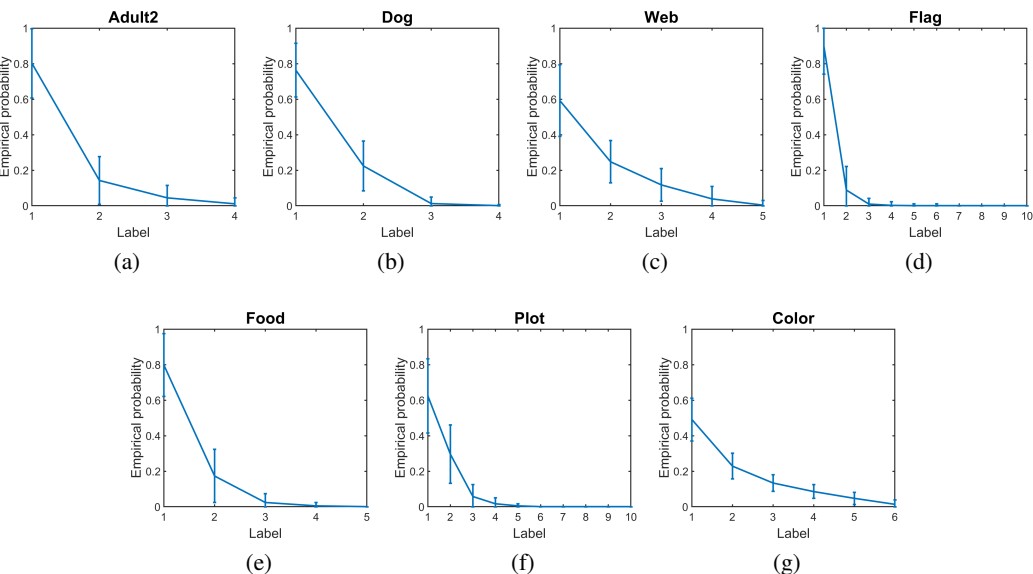

Figure 3: Empirical distribution of the mean incidence of responses sorted by the dominant proportion, averaged over all tasks in each dataset. The $i$-th data point represents the average incidence of the $i$-th highest response in each task. The error bars indicate the standard deviation of the mean incidence of the $i$-th dominating answer over the tasks in the dataset.

From the table and figure, we can observe that for all the considered public datasets the top-two answers dominate the overall answers, i.e., about 65-90% of the total answers belong to the top two. Furthermore, the average ratio from the most dominating answer to the second one is 4:1, while that between the second and the third is 7.5:1. There often exist overlaps in the error bars between

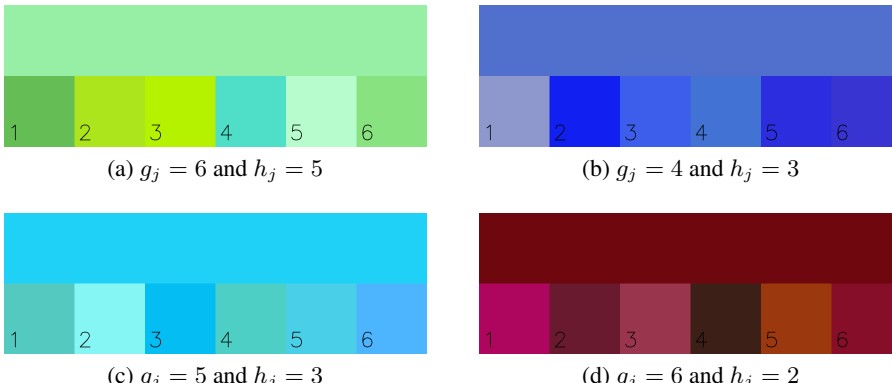

Figure 4: Example tasks for 'Color' dataset where the ground truth $g$ and the most confusing answer $h$ are determined by the color distance from the reference color (top).

the ground truth and the second dominating answer, e.g., for Web, Plot, and Color datasets, but no such overlap is found between the ground truth and the third dominating answer. What we can call a *'confusing answer'* is an answer that has an incidence rate comparable to that of the ground truth. In all the considered datasets, only the second dominating answer shows such a tendency, and thus, we can conclude that the third dominating answer cannot be called a 'confusing answer', and the top-two model in equation 1 well describes the errors in answers caused by confusion.

Moreover, from the public datasets, we also observe that the task difficulty can be quantified by the confusion probability between the top-two answers. As an example, for the Web dataset, when we select the easiest 500 tasks and hardest 500 tasks by ordering tasks with the ratio of correct answers, the ratio between the ground-truth to the 2nd best answer was 10.7:1 for the easiest group, while it was 1.5:1 for the hardest group. This observation shows that the ratio between the top-two answers indeed captures task difficulty as does our model parameter for task difficulty $q_j$ in equation 1.

## A.1 DATASETS

We collect six publicly available multi-class datasets: Adult2, Dog, Web, Flag, Food and Plot. Since these datasets do not provide information about the most confusing answer or the task difficulty, we additionally create a new dataset called 'Color', for which we can identify the most confusing answer and also quantify the task difficulty for all the included tasks.

- **Color** is a dataset where the task is to find the most similar color to the reference color among six different choices. For each task, we randomly create a reference color and then choose six choices of colors. The distance from the reference color to the ground truth color is in between 4.5 and 5.5, the distance to the most confusing answer is in between 5.5 and 6.5, and the distance to the rest of the choices is between 11 and 12, where the distance between the pairs of colors is measured by CIEDE2000 (Sharma et al., 2005) color difference formulation. The tasks are ordered in terms of their difficulty levels by measuring the gap between: the distance from the reference color to the ground truth; and that to the most confusing answer. If the distance from the reference color to the ground truth is much shorter than that to the most confusing answer, then the task is considered easy. Using MTurk, we collected 19600 labels from 196 workers for 1000 tasks. Each Human Intelligence Task (HIT) is composed of randomly selected 100 tasks, and we pay $1 to each worker who completed a HIT. Fig. 4 shows an example task for the Color dataset.

- **Adult2** (Ipeirotis et al., 2010) is a 4-class dataset where the task is to classify the web pages into four categories (G, PG, R, X) depending on the adult level of the websites. This dataset contains 3317 labels for 333 websites which are offered by 269 workers.

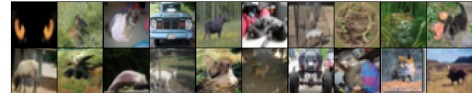 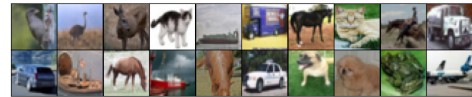

(a) Images with lowest $q$ (considered to be hard)  (b) Images with highest $q$ (considered to be easy)

Figure 5: Training images with (a) lowest and (b) highest confusion probabilities.

- **Dog** (Zhang et al., 2014) is a 4-class dataset where the task is to discriminate a breed (out of Norfolk Terrire, Norwich Terrier, Irish Wolfhound, and Scottich Deerhound) for a given dog. This dataset contains 7354 labels collected from 52 workers for 807 tasks.

- **Web** (Zhou et al., 2012) is a 5-class dataset where the task is to determine the relevance of query-URL pairs with a 5-level rating (from 1 to 5). The dataset contains 15567 labels for the 2665 query-URL pairs offered by 177 workers.

- **Flag** (Krivosheev et al., 2020) is a dataset for multiple-choice tasks where each task is to identify the country for a given flag from 10 given choices. A total of 1600 votes are collected from 220 workers for the 100 tasks.

- **Food** (Krivosheev et al., 2020) is a dataset for multiple-choice tasks where each task asks to identify a picture of a given food or dish from 5 given choices. This dataset contains 1220 labels for 76 tasks collected from 177 workers.

- **Plot** (Krivosheev et al., 2020) is a dataset for multiple-choice tasks where the task is to identify a movie from a description of its plot from 10 given choices. Only workers who correctly solved the first 10 test questions can answer the rest of the tasks. A total of 1937 labels are collected from 122 workers for 100 tasks.

Table 4 shows a summarized information for the introduced datasets.

Table 4: Dataset information

| Dataset | # workers | # tasks | # labels or choices | sparsity | $d_{task}$ | $d_{worker}$ |
|---------|-----------|---------|---------------------|----------|-----------|--------------|
| Adult2  | 269       | 333     | 4                   | 0.037    | 10.0      | 12.4         |
| Dog     | 109       | 807     | 4                   | 0.092    | 10.0      | 74.0         |
| Web     | 176       | 2653    | 5                   | 0.033    | 5.9       | 88.3         |
| Flag    | 220       | 100     | 10                  | 0.073    | 16.0      | 7.3          |
| Food    | 177       | 54      | 5                   | 0.125    | 22.1      | 6.7          |
| Plot    | 122       | 56      | 10                  | 0.293    | 35.7      | 16.4         |
| Color   | 196       | 1000    | 6                   | 0.1      | 19.5      | 99.4         |

# B    EXPERIMENTAL DETAILS FOR NEURAL NETWORK TRAINING IN SEC. 5.3

We show the details of the experiments in Sec. 5.3.

## B.1    DATASETS

The CIFAR10H dataset (Peterson et al., 2019) consists of 511,400 human classifications by 2,571 participants which were collected via Amazon Mechanical Turk. Each participant classified 200 images, 20 from each category. Every 20 tasks, a trivial question is presented to prevent random guessing, and participants who scored below 75% were excluded from the dataset. We present the images with the lowest/highest $q$ from the training samples in Fig 5. The image with a lower $q$ means that the first answer and the second answer are hard to distinguish.

## B.2 Model

We trained two simple CNN architectures, VGG-19 and ResNet-18, to show the usefulness of the second answer and the confusion probability. For each model, our loss function is defined as the cross-entropy between the softmax output and the two-hot vector (in which the values are $q$ and $1 - q$ for $g$ and $h$, respectively). We compare the results of our top-two label training with those of full-distribution training and hard label (one-hot vector) training.

## B.3 Training

We train each model using 10-fold cross validation (using 90% of images for training and 10% images for validation) and average the results across 5 runs. We run a grid search over learning rates, with the base learning rate chosen from $\{0.1, 0.01, 0.001\}$. We find 0.1 to be optimal in all cases. We trained each model for a maximum of 150 epochs using SGD optimizer with a momentum of 0.9 and a weight decay of 0.0001. Our neural networks are trained using NVIDIA GeForce 3090 GPUs.

## C BASELINE METHODS

In this section, we explain the baseline methods with which we compare the performance of our algorithms. To analyze the performance in recovering the top-two answers, we considered the ML-based algorithms, including the **Spectral-EM algorithm (MV-D&S and OPT-D&S)** (Zhang et al., 2014), **Projected Gradient Descent (PGD)** (Ma et al., 2018) and **M-MSR** (Ma & Olshevsky, 2020), which provide a "score" for each label so that we can recover the top-two answers.

- **Spectral-EM algorithm (MV-D&S and OPT-D&S)** (Zhang et al., 2014) is a two-stage algorithm for multi-class crowd labeling problems. These algorithms are built for the D&S model where each worker has his/her own confusion matrix. In the first stage of the algorithm, the confusion matrix of each worker is estimated via spectral method (OPT-D&S) or majority voting (MV-D&S), respectively, and in the second stage, the estimates for the confusion matrices are refined by optimizing the objective function of the D&S estimator via the Expectation Maximization (EM) algorithm.

- **Projected Gradient Descent (PGD)** (Ma et al., 2018) is an approach to estimate the skills of each worker in the single-coin D&S model. The authors formulate the skill estimation problem as a rank-one correlation-matrix completion problem. They propose a projected gradient descent method to solve the correlation-matrix completion problem.

- **M-MSR** (Ma & Olshevsky, 2020) algorithm is an approach to estimate the reliability of each worker in the multi-class D&S model. M-MSR algorithm utilizes that the rank of the response matrix is one. To estimate the reliability of the workers, they use update rules to find the left singular vector and right singular vector of the response matrix. In this process, the extreme values are filtered out to guarantee the stable convergence of the algorithm.

## D SYNTHETIC EXPERIMENTS

### D.1 ADDITIONAL PLOTS FOR SYNTHETIC DATA EXPERIMENTS IN SEC. 5.1

In Section 5.1, we devised four scenarios described in Table 1 to verify the robustness of our model for various $(\boldsymbol{p}, \boldsymbol{q})$ ranges, with $(n, m, s) = (50, 500, 0.2)$. The performance of algorithms is measured by the empirical average error probabilities in recovering $g_j$, $h_j$ and $(g_j, h_j)$, i.e., $\frac{1}{m} \sum_{j=1}^{m} \mathbb{P}(\hat{g}_j \neq g_j)$, $\frac{1}{m} \sum_{j=1}^{m} \mathbb{P}(\hat{h}_j \neq h_j)$ and $\frac{1}{m} \sum_{j=1}^{m} \mathbb{P}((\hat{g}_j, \hat{h}_j) \neq (g_j, h_j))$ and plotted in Fig. 6. We can observe that for all the considered scenarios TopTwo2 achieves the best performance, near the oracle MLE, in recovering $(g_j, h_j)$. Depending on scenarios though, the reason TopTwo2 outperforms can be explained differently. For Easy scenario, since $q_j$ is close to 1, it becomes easy to distinguish $g_j$ from $h_j$ but hard to distinguish $h_j$ from other labels. Our algorithm achieves the best performance in estimating $h_j$ by a large margin. For Hard scenario, it becomes hard to distinguish $g_j$ and $h_j$, but our algorithm, which uses an accurate $\hat{q}_j$, can better distinguish

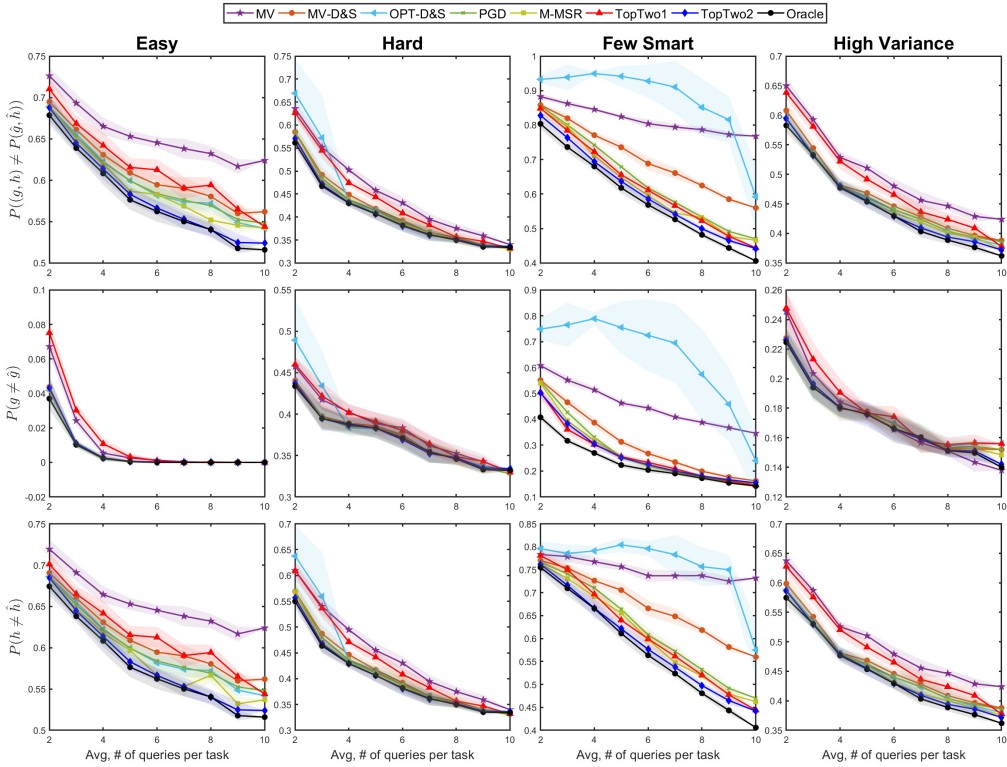

Figure 6: Prediction error for $(g_j, h_j)$ (top row), $g_j$ (middle) and $h_j$ (bottom) for four scenarios. Our algorithm (TopTwo2) achieves the best performance, near the oracle MLE for all the scenarios.

$g_j$ and $h_j$. High-variance show the effect of having diverse $q_j$ in a dataset. For Few-smart, our algorithm achieves the biggest gain compared to other methods, since our algorithm can effectively distinguish few smart workers from spammers. We remark that even though the performance gap between TopTwo2 and the next best performer is not significant for some cases, our algorithm always achieves the best performance, near that of the oracle-MLE, for all the scenarios, while the next performer keeps changing depending on scenarios. For example, the OPT D&S is the second best performer in the 'Easy' scenario, while it is the worst performer in the 'Few smart' scenario.

## D.2 ROBUSTNESS OF OUR METHODS

In this section, we present a set of four additional synthetic experiments to demonstrate the robustness of our methods, Alg. 1 and Alg. 2 (referred to as TopTwo1 and TopTwo2). In each experiment, we change a parameter of our synthetic error model and compare the prediction error of our algorithms to the baselines: majority voting(MV), OTP-D&S and MV-D&S Zhang et al. (2014), PGD Ma et al. (2018) and Oracle-MLE. We measure the performance of each algorithm by the empirical average error probabilties in recovering the ground truth $g_j$, the most confusing answer $h_j$ and the pair of top two $(g_j, h_j)$, i.e., $\frac{1}{m}\sum_{j=1}^{m}\mathbb{P}(\hat{g}_j \neq g_j)$, $\frac{1}{m}\sum_{j=1}^{m}\mathbb{P}(\hat{h}_j \neq h_j)$ and $\frac{1}{m}\sum_{j=1}^{m}\mathbb{P}((\hat{g}_j, \hat{h}_j) \neq (g_j, h_j))$. Obviously, Oracle-MLE provides a lower bound for the performance.

**Changing the dimension of observed matrix**: We first check the robustness of our methods against the change of dimensions of the observation matrix $\boldsymbol{A} \in \{0, 1 \dots, K\}^{n \times m}$ with $n \leq m$. We vary the number of workers ($n$) or the number of tasks ($m$) while fixing the other dimension. The default values of $n$ and $m$ are 50 and 500, respectively, and the sampling probability $s$ is fixed as $0.1$ throughout the experiments. The worker reliability $p_i$ and the task difficulty $q_j$ is sampled uniformly at random from $[0, 1]$ and $(1/2, 1]$, respectively, for all $i \in [n]$ and $j \in [m]$.

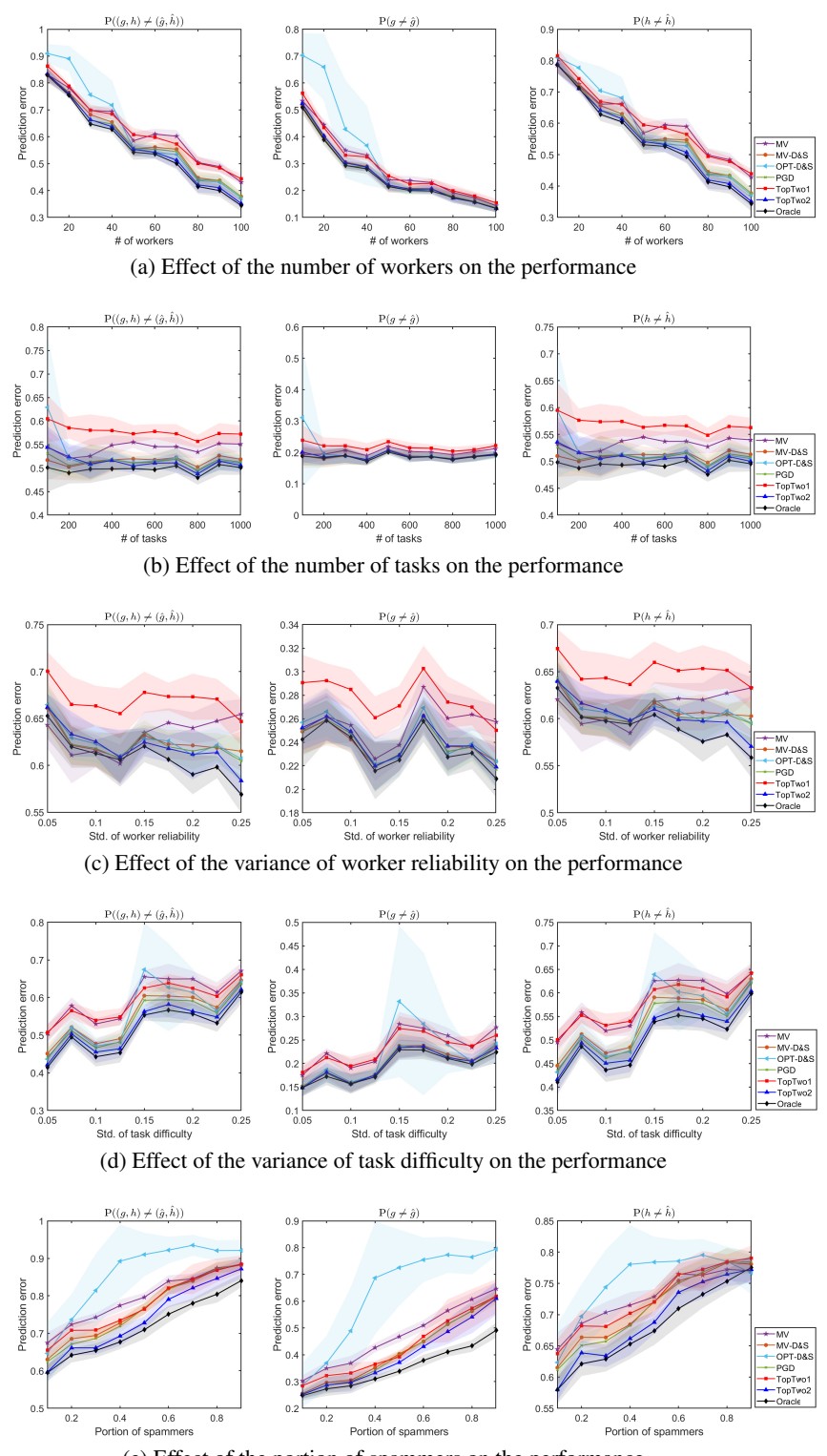

(a) Effect of the number of workers on the performance

(b) Effect of the number of tasks on the performance

(c) Effect of the variance of worker reliability on the performance

(d) Effect of the variance of task difficulty on the performance

(e) Effect of the portion of spammers on the performance

Figure 7: Prediction error for $(g_j, h_j)$ (first column), $g_j$ (second column), and $h_j$ (third column) for five different setups. The solid lines represent the mean prediction errors of each algorithm averaged over 10 runs, and the shaded regions represent the standard deviations.

In Fig. 7a and 7b, we report the results when we change $n$ for a fixed $m$ and $s$, or when we change $m$ for a fixed $n$ and $s$, respectively. From Fig. 7a, we can see that as the number of workers increases, the performance of every algorithm improves since the number of samples per task scales as $ns$ for a fixed $s$. Our algorithm achieves the performance close to the Oracle-MLE for all the considered range, which implies that the worker reliabilities $\{p_i\}$ are well estimated with our methods. From Fig. 7b, we can see that our algorithm achieves a robust performance against the change in the number of tasks, although the performance gets closer to that of Oracle-MLE as the number of tasks increases. Since our method uses SVD in the first stage, the larger dimension is beneficial for the concentration of the random perturbation matrix with respect to the expectation of the observation matrix. This phenomenon is observed for other baseline methods as well, which are based on the spectral method, OPT D&S, for example.

**Changing the variance of worker reliability**: In this experiment, we change the range of $p_i$, the parameter for worker skill/reliability, for $i \in [n]$, with a fixed mean in order to observe the impact of the variance of the worker reliability on the prediction error. We randomly sample $p_i$ from the window $[0.5 - x, 0.5 + x]$ with $x$ varying from 0.05 to 0.25. The mean of the worker reliability is fixed as 0.5.

As shown in Fig. 7c, when the variance of the worker reliability increases, the baseline methods estimating worker reliabilities perform better than the majority voting. Our TopTwo2 algorithm achieves the best performance close to Oracle-MLE, as the standard deviation increases, i.e., as the workers become more heterogeneous.

**Changing the variance of task difficulty**: We also design an experiment to observe the impact of the variance of $q_j$, $j \in [m]$, the parameter for task difficulty, on the prediction error. We randomly sample $q_j$ from the window $[0.75 - x, 0.75 + x]$ with $x$ varying from 0.05 to 0.25. The mean of the worker reliability is fixed as 0.75. If the variance of the task difficulty is small, it could be sufficient to only estimate the worker reliability since all the tasks have almost the similar task difficulties.

As shown in Fig. 7d, when the variance of the task difficulty increases, our TopTwo2 algorithm performs better than the other baselines. This is the evidence for the validity of our method in estimating the task difficulty.

**Changing the portion of spammers**: Spammers who provide random answers always exist in crowdsourcing systems. To improve the inference performance, it is important to distinguish spammers from reliable workers. In our experimental setup, we define a spammer as a worker whose reliability parameter $p_i$ is in the range $[0, 0.1]$. We change the portion of spammers among the workers from 0.1 to 0.9 and compare the prediction error of our methods to those of other baseline methods.

In Fig. 7e, we can see that our algorithm achieves the best performance among all the considered baselines except Oracle-MLE, which can exactly distinguish spammers from reliable workers. This result demonstrates the superiority of our methods in detecting spammers compared to other methods.

### D.3 ESTIMATING THE WORKER RELIABILITY VECTOR AND THE TASK DIFFICULTY VECTOR

In this section, we examine the accuracy of our estimates for the worker reliability vector $\boldsymbol{p}$ and the task difficulty vector $\boldsymbol{q}$. The worker reliability is estimated by $\hat{\boldsymbol{p}}$ defined in equation 5 of Algorithm 2 and the task difficulty is estimated by $\hat{\boldsymbol{q}}$ defined in equation 4 of Algorithm 1. To analyze the accuracy of these estimators, we compute the mean squared error (MSE), $\frac{1}{n}\|\hat{\boldsymbol{p}} - \boldsymbol{p}\|_2^2$ and $\frac{1}{m}\|\hat{\boldsymbol{q}} - \boldsymbol{q}\|_2^2$, respectively.

To analyze the estimation accuracy for the worker reliability, we first sample $p_i$ uniformly at random from $[0, 1]$ for all $i \in [n]$ and fix the worker reliability vector $\boldsymbol{p}$. Then, we randomly sample the task difficulty vector $\boldsymbol{q} \in (1/2, 1]^m$ fifty times and then sample the observation matrices from the distribution equation 1 for each $(\boldsymbol{p}, \boldsymbol{q})$ pair with a fixed $\boldsymbol{p}$. For each observation matrix, we subsample the data with varying probabilities and apply Algorithm 2 to get the estimate $\hat{\boldsymbol{p}}$, which is then used to calculate the MSE of $\boldsymbol{p}$. We report the MSE averaged over these fifty cases. Similarly, to analyze the estimation accuracy for the task difficulty, we randomly sample and fix a task difficulty vector $\boldsymbol{q} \in (1/2, 1]^m$ and generate fifty different observation matrices while varying the worker reliability vector $\boldsymbol{p}$. We again report the MSE averaged over these fifty cases. The number of workers

and that of tasks is set to be $(50, 500)$ for the worker reliability estimation, and to be $(100, 1000)$ for the task difficulty estimation.

In Fig. 8a and 8b, we plot the MSE for $\boldsymbol{p}$ and $\boldsymbol{q}$, respectively, as the average number of queries per task increases. We can see that both for $\boldsymbol{p}$ and $\boldsymbol{q}$, the MSEs converge to near zero as the average number of queries per task increases. However, estimating the task difficulty requires more number of samples as our theory equation 11 suggests.

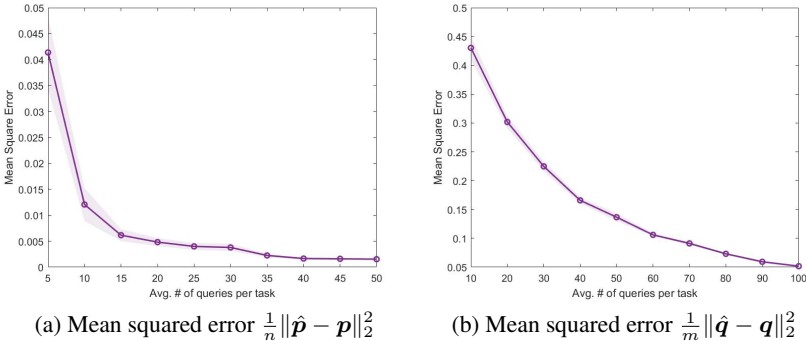

(a) Mean squared error $\frac{1}{n}\|\hat{\boldsymbol{p}} - \boldsymbol{p}\|_2^2$        (b) Mean squared error $\frac{1}{m}\|\hat{\boldsymbol{q}} - \boldsymbol{q}\|_2^2$

Figure 8: Mean squared errors in estimating the worker reliability vector $\boldsymbol{p}$ (left) and the task difficulty vector $\boldsymbol{q}$ (right), respectively.

## E  DISCUSSION OF THEORETICAL RESULTS

In this section, we present a discussion of the main theoretical results.

- Theorem 1 asserts that the sampling probability of $\Omega\left(\frac{1}{\delta_1^2 \|\boldsymbol{p}\|_2^2} \log \frac{K}{\epsilon}\right)$ is sufficient to recover the top-two answers $(g_j, h_j)$ for any task $j \in [m]$ and to estimate the confusion probability $q_j$ with accuracy of $|\hat{q}_j - q_j| < \delta_1$ by Algorithm 1 with probability at least $1 - \epsilon$. Combined with Theorem 3 part (a), we can see that this sample complexity is the minimax optimal rate for a fixed collective quality of workers, measured by $\|\boldsymbol{p}\|_2^2$.

- It is also worth comparing our algorithm with the simple majority voting (MV) scheme. The MV scheme infers the top-two answers by counting the majority of the received answers. Simple analysis shows that the MV scheme requires the sampling probability $s$ such that $ns = \Theta\left((\frac{1}{n}\sum_i p_i)^{-2} \log \frac{1}{\epsilon}\right)$ to recover $(g_j, h_j)$ with probability $1 - \epsilon$. Remind that Algorithm 1 requires $ns = \Omega\left(\frac{n}{\delta_1^2 \|\boldsymbol{p}\|_2^2} \log \frac{K}{\epsilon}\right)$ samples per task. Since $\frac{1}{n}\|\boldsymbol{p}\|^2 = \frac{1}{n}\sum_i p_i^2 \geq \left(\frac{1}{n}\sum_i p_i\right)^2$ by Cauchy-Schwarz inequality, Algorithm 1 achieves strictly better trade-offs unless $p_i$ is same for all workers $i \in [n]$. As an example, for a spammer-hammer model where $\alpha \in (0, 1)$ fraction of workers are hammers with $p_i = 1$ and the rest are spammers with $p_i = 0$, Algorithm 1 requires $ns = \Theta\left(\frac{1}{\alpha} \log \frac{1}{\epsilon}\right)$ samples per task, while MV requires $ns = \Theta\left(\frac{1}{\alpha^2} \log \frac{1}{\epsilon}\right)$ samples per task to recover top-two answers with probability $1 - \epsilon$.

- Theorem 2 shows that when we have an entrywise bound on the estimated worker reliability vector $\boldsymbol{p}$ and the task difficulty vector $\boldsymbol{q}$, the plug-in MLE estimator, used in Algorithm 2, guarantees the recovery of top-two answers if the sampling probability $s = \Omega(\frac{\log(m/\epsilon)}{n\bar{D}})$ where $\bar{D}$, which depend on $(\boldsymbol{p}, \boldsymbol{q})$, indicates the average reliability of workers in distinguishing the top-two answers from any other pairs for the most difficult task. Combined with Theorem 3 part (b), we can see that this sample complexity is the minimax optimal rate for any $(\boldsymbol{p}, \boldsymbol{q})$, ignoring the logarithmic terms.

- Combining the conditions for the accurate estimation of model parameters in equation 11 and the convergence of the plug-in MLE (Theorem 2), Corollary 2 shows the condition on the sample complexity to guarantee the performance of Algorithm 2.

# F    PROOF OF PROPOSITION 1

For each task $j$ and label $k$, define four indicator functions:

$$
\begin{aligned}
\Pi_a(j,k) &:= \mathbb{1}(g_j > k, h_j > k), \\
\Pi_b(j,k) &:= \mathbb{1}(g_j \le k, h_j > k), \\
\Pi_c(j,k) &:= \mathbb{1}(g_j > k, h_j \le k), \\
\Pi_d(j,k) &:= \mathbb{1}(g_j \le k, h_j \le k),
\end{aligned}
\tag{17}
$$

which satisfy $\Pi_a(j,k) + \Pi_b(j,k) + \Pi_c(j,k) + \Pi_d(j,k) = 1$. For notational simplicity, we will often drop $(j,k)$ fron $\Pi_*$. The pmf of $\boldsymbol{A}^{(k)}$ is given by

$$
A_{ij}^{(k)} = \begin{cases} -1 & \text{with probability } s(1 - \rho_{ij}^{(k)}), \\ 1 & \text{with probability } s\rho_{ij}^{(k)}, \\ 0 & \text{with probability } 1 - s, \end{cases}
\tag{18}
$$

where $\rho_{ij}^{(k)} = \Pi_a(j,k)p_i + \Pi_b(j,k)p_i(1-q_j) + \Pi_c(j,k)p_iq_j + \frac{(K-k)(1-p_i)}{K}$, and its expectation is $\mathbb{E}[A_{ij}^{(k)}] = s(2\rho_{ij}^{(k)} - 1)$. Note that by using $\Pi_a = 1 - \Pi_b - \Pi_c - \Pi_d$, the probability $\rho_{ij}^{(k)}$ can be written as $\rho_{ij}^{(k)} = p_i\left(q_j(\Pi_c - \Pi_b) - (\Pi_c + \Pi_d) + \frac{k}{K}\right) + \frac{K-k}{K}$. Thus, by defining

$$
r_j^{(k)} := q_j(\Pi_c - \Pi_b) - (\Pi_c + \Pi_d) + \frac{k}{K},
\tag{19}
$$

the expectation of $A_{ij}^{(k)}$ can be written as

$$
\mathbb{E}[A_{ij}^{(k)}] = s(2\rho_{ij}^{(k)} - 1) = s\left(2p_ir_j^{(k)} + \frac{K-2k}{K}\right),
\tag{20}
$$

and

$$
\mathbb{E}[\boldsymbol{A}^{(k)}] - \frac{s(K-2k)}{K}\mathbb{1}_{n \times m} = 2s\boldsymbol{p}(\boldsymbol{r}^{(k)})^\top.
\tag{21}
$$

Note that

$$
\begin{array}{ll}
\text{Case I: } g_j > h_j & \text{Case II: } g_j < h_j \\
\Pi_a(j,k) = 1 \text{ where } k < h_j, & \Pi_a(j,k) = 1 \text{ where } k < g_j, \\
\Pi_c(j,k) = 1 \text{ where } h_j \le k < g_j, & \Pi_b(j,k) = 1 \text{ where } g_j \le k < h_j, \\
\Pi_d(j,k) = 1 \text{ where } g_j \le k; & \Pi_d(j,k) = 1 \text{ where } h_j \le k.
\end{array}
\tag{22}
$$

Thus, $r_j^{(k)}$ in equation 19 is equal to

$$
\text{Case I: } g_j > h_j \qquad\qquad\qquad \text{Case II: } g_j < h_j
$$

$$
r_j^{(k)} = \begin{cases} \frac{k}{K} & \text{where } k < h_j; \\ \frac{k}{K} - (1-q_j) & \text{where } h_j \le k < g_j; \\ \frac{k}{K} - 1 & \text{where } g_j \le k, \end{cases}
\qquad
r_j^{(k)} = \begin{cases} \frac{k}{K} & \text{where } k < g_j; \\ \frac{k}{K} - q_j & \text{where } g_j \le k < h_j; \\ \frac{k}{K} - 1 & \text{where } h_j \le k. \end{cases}
$$

# G    PERFORMANCE ANALYSIS OF ALGORITHM 1

## G.1    PROOFS OF THEOREM 1 AND COROLLARY 1

In Algorithm 1, we use the data matrix $\boldsymbol{A}^1$, which is obtained by randomly splitting the original data matrix $\boldsymbol{A}$ into $\boldsymbol{A}^1$ and $\boldsymbol{A}^2$ with probability $s_1$ and $(1 - s_1)$, respectively. Then, the first stage of Algorithm 1 begins with randomly splitting $\boldsymbol{A}^1$ again into two independent matrices $\boldsymbol{B}$ and $\boldsymbol{C}$ with equal probabilities, and then converting $\boldsymbol{B}$ and $\boldsymbol{C}$ into $(K-1)$-binary matrices $\boldsymbol{B}^{(k)}$ and $\boldsymbol{C}^{(k)}$ as explained in Sec. 2. We define $\boldsymbol{X}^{(k)}$ and $\boldsymbol{Y}^{(k)}$ as $\boldsymbol{X}^{(k)} := \boldsymbol{B}^{(k)} - \frac{s'(K-2k)}{K}\mathbb{1}_{n \times m}$ and $\boldsymbol{Y}^{(k)} := \boldsymbol{C}^{(k)} - \frac{s'(K-2k)}{K}\mathbb{1}_{n \times m}$ where $s' = s \cdot s_1/2$. We have $\mathbb{E}[\boldsymbol{X}^{(k)}] = \mathbb{E}[\boldsymbol{Y}^{(k)}] = s'\boldsymbol{p}(\boldsymbol{r}^{(k)})^\top$

from Prop. 1. For notational simplicity, we will ignore this random splitting for a moment and just pretend that $\boldsymbol{X}^{(k)}$ and $\boldsymbol{Y}^{(k)}$ are sampled independently with $s' = s$ throughout this section.

We first outline the proof. Based on the observation that $\mathbb{E}[\boldsymbol{X}^{(k)}] = s\boldsymbol{p}(\boldsymbol{r}^{(k)})^\top$, if $\mathbb{E}[\boldsymbol{X}^{(k)}]$ is available we can recover $\boldsymbol{p}^* = \frac{\boldsymbol{p}}{\|\boldsymbol{p}\|_2}$ by SVD, and by using $\boldsymbol{p}^*$ it is possible to recover $\|\boldsymbol{p}\|_2 \boldsymbol{r}^{(k)}$, which then reveals $\{(g_j, h_j)\}_{j=1}^m$ as well as $\boldsymbol{q}$ from the relation in equation 2. To estimate $\boldsymbol{p}^*$ from $\boldsymbol{X}^{(k)}$, we first bound the spectral norm of the perturbation, $\|\boldsymbol{X}^{(k)} - \mathbb{E}[\boldsymbol{X}^{(k)}]\|_2$. We then use this bound and Wedin SinΘ theorem to bound $\sin\theta(\boldsymbol{u}^{(k)}, \boldsymbol{p}^*)$ where $\boldsymbol{u}^{(k)}$ is the left singular vector of $\boldsymbol{X}^{(k)}$ with the largest singular value. We trim the abnormally large components of $\boldsymbol{u}^{(k)}$ and denote the resulting vector by $\tilde{\boldsymbol{u}}^{(k)}$. After trimming, it is still possible to show that $\sin\theta(\tilde{\boldsymbol{u}}^{(k)}, \boldsymbol{p}^*)$ can be bounded in the same order as that of $\sin\theta(\boldsymbol{u}^{(k)}, \boldsymbol{p}^*)$. Finally, we provide an entrywise bound between $\boldsymbol{v}^{(k)} = \frac{2}{s}(\boldsymbol{Y}^{(k)})^\top \tilde{\boldsymbol{u}}^{(k)}$ and $\|\boldsymbol{p}\|_2 \boldsymbol{r}^{(k)}$ in Lemma 5, which is the main lemma to prove Theorem 1. We state our main technical lemmas first and then prove Theorem 1.

Let us define the perturbation matrix

$$\boldsymbol{E} := \boldsymbol{X}^{(k)} - \mathbb{E}[\boldsymbol{X}^{(k)}] = \boldsymbol{B}^{(k)} - \frac{s(K - 2k)}{K}\mathbb{1}_{n \times m} - s\boldsymbol{p}(\boldsymbol{r}^{(k)})^\top = \boldsymbol{B}^{(k)} - \mathbb{E}[\boldsymbol{B}^{(k)}] \quad (23)$$

where

$$B_{ij}^{(k)} = \begin{cases} -1 & \text{w.p. } s(1 - \rho_{ij}^{(k)}), \\ 1 & \text{w.p. } s\rho_{ij}^{(k)}, \\ 0 & \text{w.p. } 1 - s, \end{cases} \quad (24)$$

and $\rho_{ij}^{(k)} = \Pi_a(j, k)p_i + \Pi_b(j, k)p_i(1 - q_j) + \Pi_c(j, k)p_i q_j + \frac{(K-k)(1-p_i)}{K}$ for $(\Pi_a, \Pi_b, \Pi_c, \Pi_d)$ defined in equation 17.

For the perturbation matrix $\boldsymbol{E}$ in equation 23, we have

$$\mathbb{E}[E_{i,j}] = 0, \quad \text{and} \quad |E_{i,j}| \le 2, \quad 1 \le i \le n, \ 1 \le j \le m, \quad (25)$$

and also

$$\begin{aligned} \text{var}(E_{ij}) = \text{var}(B_{ij}^{(k)}) &= \mathbb{E}[(B_{ij}^{(k)})^2] - (\mathbb{E}[B_{ij}^{(k)}])^2 \\ &= s - (s(\rho_{ij}^{(k)} - 1/2))^2 \le s. \end{aligned} \quad (26)$$

Note that $\{E_{ij}\}$ are independent across all $i, j$. Define

$$\nu := \max\left\{ \max_i \sum_j \mathbb{E}[E_{i,j}^2], \ \max_j \sum_i \mathbb{E}[E_{i,j}^2] \right\} \le \max\{m, n\}s. \quad (27)$$

By applying the spectral norm bound to random matrices with independent entires, appeared in Bandeira & Van Handel (2016) and summarized in Theorem 4, we can bound the spectral norm of $\boldsymbol{E}$ as below.

**Lemma 2** (Spectral norm bound of $\boldsymbol{E}$). *With probability* $1 - (n + m)^{-8}$, *we have*

$$\|\boldsymbol{E}\| \le 4\sqrt{s\max(m, n)} + \tilde{c}\sqrt{\log(n + m)} \quad (28)$$

*for some constant* $\tilde{c} > 0$ *when* $m \ge n$. *For some sufficiently large* $m$, *assuming* $n = o(m)$ *and* $s = \Omega(\log(n + m)/m)$, *the spectral norm of* $\boldsymbol{E}$ *can be further bounded by*

$$\|\boldsymbol{E}\| \le 5\sqrt{sm}. \quad (29)$$

Using the bounded spectral norm of $\boldsymbol{E}$ in equation 29 and applying the Wedin SinΘ theorem, summarized in Theorem 5, we can bound the angle between $\boldsymbol{u}^{(k)}$ and $\boldsymbol{p}^*$.

**Lemma 3.** *For some sufficiently large* $m$, *assuming* $n = o(m)$ *and* $s = \Omega(\log(n + m)/m)$, *we have*

$$\sin\theta(\boldsymbol{u}^{(k)}, \boldsymbol{p}^*) \le \Theta(1/\sqrt{sn}) \quad (30)$$

*with probability at least* $1 - (n + m)^{-8}$.

*Proof.* By applying the Wedin SinΘ Theorem (Theorem 5), we have

$$\sin\theta(\boldsymbol{u}^{(k)},\boldsymbol{p}^*) \leq \frac{\sqrt{2}\|\boldsymbol{E}\|}{s\|\boldsymbol{p}\|_2 \cdot \|\boldsymbol{r}^{(k)}\|_2 - \|\boldsymbol{E}\|}. \tag{31}$$

We have $\|\boldsymbol{p}\|_2 = \Theta(\sqrt{n})$ and $\|\boldsymbol{r}^{(k)}\|_2 = \Theta(\sqrt{m})$ by assumptions on model parameters. By Lemma 2, for some sufficiently large $m$, assuming $n = o(m)$ and $s = \Omega(\log(n+m)/m)$, we have $\|\boldsymbol{E}\| \leq 5\sqrt{sm}$ with probability at least $1 - (n+m)^{-8}$. Combining these bounds, we get

$$\sin\theta(\boldsymbol{u}^{(k)},\boldsymbol{p}^*) \leq \frac{\Theta(\sqrt{sm})}{\Theta(s\sqrt{mn}) - \Theta(\sqrt{sm})} = \frac{1}{\Theta(\sqrt{sn})}. \tag{32}$$

$\square$

We trim the abnormally large components of $\boldsymbol{u}^{(k)}$ by letting it zero if $u_i^{(k)} > 2/(\eta\sqrt{n})$ and denote the resulting vector as $\tilde{\boldsymbol{u}}^{(k)}$. This process is required to control the maximum entry size of $\tilde{\boldsymbol{u}}^{(k)}$, which is used later in the proof. For the next lemma, we show that after the trimming process, the norm of $\tilde{\boldsymbol{u}}^{(k)}$ is still close to 1 and the angle between $\tilde{\boldsymbol{u}}^{(k)}$ and $\boldsymbol{p}^*$ has the same order as that of $\sin\theta(\boldsymbol{u}^{(k)},\boldsymbol{p}^*)$.

**Lemma 4.** *Given $\|\boldsymbol{p}^*\|_2 \geq \eta\sqrt{n}$, we have*

$$\|\tilde{\boldsymbol{u}}^{(k)}\|_2 \geq \sqrt{1 - 50\sin^2\theta(\boldsymbol{u}^{(k)},\boldsymbol{p}^*)}, \tag{33}$$

$$\sin\theta(\tilde{\boldsymbol{u}}^{(k)},\boldsymbol{p}^*) \leq 6\sqrt{2}\sin\theta(\boldsymbol{u}^{(k)},\boldsymbol{p}^*). \tag{34}$$

The proof of Lemma 4 is provided in Section G.2.

Finally, we provide our main lemma giving the entrywise bound on the difference between $\boldsymbol{v}^{(k)} = \frac{1}{s}(\boldsymbol{Y}^{(k)})^\top\tilde{\boldsymbol{u}}^{(k)}$ and $\|\boldsymbol{p}\|_2\boldsymbol{r}^{(k)}$.

**Lemma 5** (Entrywise Bound). *For any $\delta_1, \epsilon > 0$, and any task $j \in [m]$ and label index $k \in [K]$, if the sampling probability $s \geq \Theta\left(\frac{1}{\delta_1^2\|\boldsymbol{p}\|_2^2}\log\frac{1}{\epsilon}\right)$, then we can guarantee*

$$\mathbb{P}\left(\left|\frac{1}{s}\left\langle\boldsymbol{Y}_{*j}^{(k)},\tilde{\boldsymbol{u}}^{(k)}\right\rangle - \|\boldsymbol{p}\|_2 r_j^{(k)}\right| < \delta_1\|\boldsymbol{p}\|_2\right) > 1 - \epsilon \tag{35}$$

*as $m \to \infty$ when $n = O(m/\log m)$.*

*Proof.* For notional simplicity, denote $\theta(\tilde{\boldsymbol{u}}^{(k)},\boldsymbol{p}^*)$ by $\theta$. To prove equation 35, we show bounds on two probabilities,

$$\mathbb{P}\left(\left|\frac{1}{s}\left\langle\boldsymbol{Y}_{*j}^{(k)},\tilde{\boldsymbol{u}}^{(k)}\right\rangle - \|\tilde{\boldsymbol{u}}^{(k)}\|_2\|\boldsymbol{p}\|_2 r_j^{(k)}\cos\theta\right| > \frac{\delta_1\|\boldsymbol{p}\|_2}{2}\right) < \epsilon/2, \tag{36}$$

$$\mathbb{P}\left(\left|\|\tilde{\boldsymbol{u}}^{(k)}\|_2\|\boldsymbol{p}\|_2 r_j^{(k)}\cos\theta - \|\boldsymbol{p}\|_2 r_j^{(k)}\right| > \frac{\delta_1\|\boldsymbol{p}\|_2}{2}\right) < \epsilon/2. \tag{37}$$

Then, the triangle inequality implies equation 35.

We first prove equation 36. Remind that we do the random splitting of the input matrix $\boldsymbol{A}$ and define the two independent binary-converted matrices as $\boldsymbol{X}^{(k)}$ and $\boldsymbol{Y}^{(k)}$, for $1 \leq k < K$, which are used to estimate $\tilde{\boldsymbol{u}}^{(k)}$ and $\boldsymbol{v}^{(k)}$, respectively. Thus, $\tilde{\boldsymbol{u}}^{(k)}$ is independent from $\boldsymbol{Y}^{(k)}$ and this independence is used when we bound the first and second moments of $v_j^{(k)} = \frac{1}{s}\langle\boldsymbol{Y}_{*j}^{(k)},\tilde{\boldsymbol{u}}^{(k)}\rangle$. For any $1 \leq j \leq m$, the first and second moments of $v_j^{(k)} = \frac{1}{s}\langle\boldsymbol{Y}_{*j}^{(k)},\tilde{\boldsymbol{u}}^{(k)}\rangle$ satisfy

$$\mathbb{E}\left[\frac{1}{s}\left\langle\boldsymbol{Y}_{*j}^{(k)},\tilde{\boldsymbol{u}}^{(k)}\right\rangle\right] = \langle\boldsymbol{p},\tilde{\boldsymbol{u}}^{(k)}\rangle r_j^{(k)} = \|\boldsymbol{p}\|_2\|\tilde{\boldsymbol{u}}^{(k)}\|_2(\cos\theta)r_j^{(k)} = \Theta(\sqrt{n}) \tag{38}$$

if $r_j^{(k)} \neq 0$ by Lemma 3 and 4, and

$$\text{var}\left(\frac{1}{s}\left\langle\boldsymbol{Y}_{*j}^{(k)},\tilde{\boldsymbol{u}}^{(k)}\right\rangle\right) \leq \frac{1}{s^2}\sum_{i=1}^n(\tilde{u}_i^{(k)})^2\mathbb{E}[(Y_{ij}^{(k)})^2] = \Theta\left(\frac{1}{s}\right) \tag{39}$$

since $\mathbb{E}[(Y_{ij}^{(k)})^2] = \Theta(s)$ and $\sum_{i=1}^n (\tilde{u}_i^{(k)})^2 = \Theta(1)$ by Lemma 3 and 4. Furthermore, we have $\max_{1 \leq i \leq m} |Y_{ij}^{(k)} \tilde{u}_i^{(k)}| \leq \Theta\left(\frac{1}{\sqrt{n}}\right)$ since $\tilde{u}_i^{(k)} \leq \frac{2}{\eta\sqrt{n}}$. By applying the Bernstein's inequality, we can show that

$$\mathbb{P}\left(\left|\frac{1}{s}\left\langle \boldsymbol{Y}_{*j}^{(k)}, \tilde{\boldsymbol{u}}^{(k)} \right\rangle - \|\tilde{\boldsymbol{u}}^{(k)}\|_2 \|\boldsymbol{p}\|_2 r_j^{(k)} \cos\theta\right| > \frac{\delta_1 \|\boldsymbol{p}\|_2}{2}\right) \leq 2\exp\left(-\frac{\Theta(\delta_1^2 \|\boldsymbol{p}\|_2^2)}{\Theta\left(\frac{1}{s}\right) + \Theta\left(\delta_1 \|\boldsymbol{p}\|_2/\sqrt{n}\right)}\right)$$
$$\leq \exp\left(-\Theta(s\delta_1^2 \|\boldsymbol{p}\|_2^2)\right) \tag{40}$$

where the second inequality is due to the assumption $\|\boldsymbol{p}\|_2 = \Theta(\sqrt{n})$. To make this probability less than $\frac{\epsilon}{2}$, it is sufficient to have $s \geq \Omega\left(\frac{1}{\delta_1^2 \|\boldsymbol{p}\|_2^2} \log\frac{1}{\epsilon}\right)$.

We next prove equation 37 by bounding $\left|\|\tilde{\boldsymbol{u}}^{(k)}\|_2 \|\boldsymbol{p}\|_2 r_j^{(k)} \cos\theta - \|\boldsymbol{p}\|_2 r_j^{(k)}\right|$. By the triangle inequality, we have

$$\left|\|\tilde{\boldsymbol{u}}^{(k)}\|_2 \|\boldsymbol{p}\|_2 r_j^{(k)} \cos\theta - \|\boldsymbol{p}\|_2 r_j^{(k)}\right| \leq \left|\|\tilde{\boldsymbol{u}}^{(k)}\|_2 \|\boldsymbol{p}\|_2 r_j^{(k)} \cos\theta - \|\boldsymbol{p}\|_2 r_j^{(k)} \cos\theta\right|$$
$$+ \left|\|\boldsymbol{p}\|_2 r_j^{(k)} \cos\theta - \|\boldsymbol{p}\|_2 r_j^{(k)}\right|. \tag{41}$$

Note that

$$\frac{1}{\|\boldsymbol{p}\|_2} \cdot \left|\|\tilde{\boldsymbol{u}}^{(k)}\|_2 \|\boldsymbol{p}\|_2 r_j^{(k)} \cos\theta - \|\boldsymbol{p}\|_2 r_j^{(k)} \cos\theta\right| = r_j^{(k)} \cos\theta \left|\|\tilde{\boldsymbol{u}}^{(k)}\|_2 - 1\right|$$
$$\leq \Theta(\sin^2\theta(\boldsymbol{u}^{(k)}, \boldsymbol{p}^*)) = \frac{1}{\Theta(ns)}, \tag{42}$$

with probability $1 - (n+m)^{-8}$ by Lemma 3 and 4, and also note that

$$\frac{1}{\|\boldsymbol{p}\|_2} \cdot \left|\|\boldsymbol{p}\|_2 r_j^{(k)} \cos\theta - \|\boldsymbol{p}\|_2 r_j^{(k)}\right| = r_j^{(k)}(1 - \cos\theta)$$
$$\leq \Theta(\sin^2\theta(\boldsymbol{u}^{(k)}, \boldsymbol{p}^*)) = \frac{1}{\Theta(ns)}, \tag{43}$$

with probability $1 - (n+m)^{-8}$ by Lemma 3 and 4. To make these errors of order $1/\Theta(ns)$ less than $\frac{\delta_1}{2}$, it is sufficient to have $s \geq \Omega\left(\frac{1}{\delta_1 n}\right)$.

By combining the above results, it can be guaranteed that $\left|\frac{1}{2s}\left\langle \boldsymbol{Y}_{*j}^{(k)}, \tilde{\boldsymbol{u}}^{(k)} \right\rangle - \|\boldsymbol{p}\|_2 r_j^{(k)}\right| < \delta\|\boldsymbol{p}\|_2$ with probability at least $1 - \epsilon$, if the sampling probability

$$s \geq \max\left\{\Omega\left(\frac{1}{\delta_1^2 \|\boldsymbol{p}\|_2^2} \log\frac{1}{\epsilon}\right), \Omega\left(\frac{1}{\delta_1 n}\right)\right\} = \Omega\left(\frac{1}{\delta_1^2 \|\boldsymbol{p}\|_2^2} \log\frac{1}{\epsilon}\right) \tag{44}$$

where the last equality is due to $\|\boldsymbol{p}\|_2 = \Theta(\sqrt{n})$. The condition $s = \Omega(\log(n+m)/m)$ in Lemma 3 is immediately satisfied by equation 44 when $n = O(m/\log m)$. $\qquad\square$

**Proof of Theorem 1.** By using Lemma 5, we next prove Theorem 1. By applying the union bound over $k \in [K]$, if $s \geq \Theta\left(\frac{1}{\delta_1^2 \|\boldsymbol{p}\|_2^2} \log\frac{K}{\epsilon}\right)$ then we have

$$\|\boldsymbol{p}\|_2(r_j^{(k)} - \delta_1) \leq v_j^{(k)} = \frac{1}{s}\left\langle \boldsymbol{Y}_{*j}^{(k)}, \tilde{\boldsymbol{u}}^{(k)} \right\rangle \leq \|\boldsymbol{p}\|_2(r_j^{(k)} + \delta_1), \quad \forall k \in [K] \tag{45}$$

for any $\delta_1 > 0$ and $j \in [m]$ with probability at least $1 - \epsilon$. Under the condition equation 45, for any $q_j \in (1/2, 1)$ and $\delta < \min\left\{\frac{2q_j - 1}{2}, \frac{1 - q_j}{2}\right\}$, we can guarantee that

$$\frac{1}{K} - q_j + \delta < \frac{1}{K} - (1 - q_j) - \delta \quad \text{and} \quad \frac{1}{K} - (1 - q_j) + \delta < \frac{1}{K} - \delta, \tag{46}$$

which implies $(\hat{g}_j, \hat{h}_j) = (g_j, h_j)$ for $(\hat{g}_j, \hat{h}_j)$ defined in equation 3. This proves equation 8 of Theorem 1.

We next prove equation 9, the accuracy guarantee in estimating the task difficulty vector $\boldsymbol{q}$. After estimating $\|\boldsymbol{p}\|_2 \boldsymbol{r}^{(k)}$ by $\boldsymbol{v}^{(k)} = \frac{1}{s}(\boldsymbol{Y}^{(k)})^\top \tilde{\boldsymbol{u}}^{(k)}$, we estimate $\|\boldsymbol{p}\|_2$ by calculating $l$ where $l_j := \frac{K}{K-2} \sum_{k \neq \hat{g}_j, k \neq \hat{h}_j} \Delta v_j^{(k)}$ and $l := \frac{1}{m} \sum_{j=1}^m l_j$. Assume that $|\|\boldsymbol{p}\|_2 - l| \leq \|\boldsymbol{p}\|_2 \delta'$. We will specify the required order of $\delta'$ later. Remind that the estimate for $q_j$ is defined as $\hat{q}_j := \frac{1}{K} - \frac{\Delta v_j^{(\hat{g}_j)}}{l}$. Under the condition that $\hat{g}_j = g_j$ and $|v_j - \|\boldsymbol{p}\|_2 r_j^{(k)}| \leq \|\boldsymbol{p}\|_2 \delta_1$, both of which are satisfied under the conditions of Lemma 5, we have

$$\frac{\left(\frac{1}{K} - q_j - 2\delta_1\right)}{1 + \delta'} \leq \frac{\Delta v_j^{(\hat{g}_j)}}{l} \leq \frac{\left(\frac{1}{K} - q_j + 2\delta_1\right)}{1 - \delta'}. \tag{47}$$

By the Taylor expansion for $\frac{1}{1-x} = 1 + x + \Theta(x^2)$ as $x \to 0$, we have

$$|\hat{q}_j - q_j| \leq 2\delta_1 + \delta'\left(\frac{1}{K} - q_j + 2\delta_1\right) + \Theta(\delta'^2) = \Theta(\delta_1 + \delta'). \tag{48}$$

Thus, both the order of $\delta'$, which is the estimation error of $\|\boldsymbol{p}\|_2$, and that of $\delta$, which is the estimation error of $\|\boldsymbol{p}\|_2 r_j^{(k)}$, govern the estimation accuracy of $q_j$. We next show that we can have $\delta' = \Theta(\delta_1)$. By Lemma 5, we have $|v_j - \|\boldsymbol{p}\|_2 r_j^{(k)}| \leq \|\boldsymbol{p}\|_2 \delta_1$, which implies

$$\|\boldsymbol{p}\|_2(\Delta r_j^{(k)} - 2\delta_1) \leq \Delta v_j^{(k)} \leq \|\boldsymbol{p}\|_2(\Delta r_j^{(k)} + 2\delta_1). \tag{49}$$

Under the condition $(\hat{g}_j, \hat{h}_j) = (g_j, h_j)$, since $\Delta r_j^{(k)} = \frac{1}{K}$ for $k \neq \hat{g}_j, \hat{h}_j$, we have

$$\|\boldsymbol{p}\|_2 - \|\boldsymbol{p}\|_2 \frac{2\delta_1 K}{K-2} \leq l_j = \frac{K}{K-2} \sum_{k \neq \hat{g}_j, k \neq \hat{h}_j} \Delta v_j^{(k)} \leq \|\boldsymbol{p}\|_2 + \|\boldsymbol{p}\|_2 \frac{2\delta_1 K}{K-2}, \tag{50}$$

and thus $\delta' = \frac{2\delta_1 K}{K-2} = \Theta(\delta_1)$. Thus, it is enough to have $s = \Omega\left(\frac{1}{\delta_1^2 \|\boldsymbol{p}\|_2^2} \log \frac{K}{\epsilon}\right)$ to guarantee equation 9.

**Proof of Corollary 1.** By using Lemma 5 and taking the union bound over all tasks $j \in [m]$ as well as $k \in [K]$, we can prove Corollary 1 in a similar way as that of Theorem 1.

### G.2 Proof of Lemma 4

We first prove equation 33,

$$\|\tilde{\boldsymbol{u}}^{(k)}\|_2 \geq \sqrt{1 - 50\sin^2\theta(\boldsymbol{u}^{(k)}, \boldsymbol{p}^*)}.$$

Let $I$ be the set of indices $1 \leq i \leq n$ such that $u_i^{(k)} \geq \frac{2}{\eta\sqrt{n}}$. Then, we have $u_i^{(k)} - p_i^* \geq \frac{1}{\eta\sqrt{n}}$ for all $i \in I$ since $p_i^* = p_i/\|\boldsymbol{p}\|_2 \leq \frac{1}{\eta\sqrt{n}}$ due to the assumption that $\|\boldsymbol{p}\|_2^2 \geq \eta^2 n$. Thus, we have

$$\frac{|I|}{\eta^2 n} \leq \sum_{i \in I}(u_i^{(k)} - p_i^*)^2 \leq \|\boldsymbol{u}^{(k)} - \boldsymbol{p}^*\|_2^2. \tag{51}$$

By using the triangle inequality, we can show that

$$
\begin{aligned}
\sqrt{\sum_{i \in I}\left(u_i^{(k)}\right)^2} &\leq \sqrt{\sum_{i \in I}\left(u_i^{(k)} - \frac{2}{\eta\sqrt{n}}\right)^2} + \sqrt{\frac{4|I|}{\eta^2 n}} \\
&\leq \sqrt{\sum_{i \in I}\left(p_i^* - \frac{2}{\eta\sqrt{n}}\right)^2} + \sqrt{\sum_{i \in I}\left(u_i^{(k)} - p_i^*\right)^2} + \sqrt{\frac{4|I|}{\eta^2 n}} \\
&\leq \sqrt{\frac{4|I|}{\eta^2 n}} + \sqrt{\sum_{i \in I}\left(u_i^{(k)} - p_i^*\right)^2} + \sqrt{\frac{4|I|}{\eta^2 n}} \\
&\leq 5\|\boldsymbol{u}^{(k)} - \boldsymbol{p}^*\|_2.
\end{aligned}
\tag{52}
$$

Therefore, we get

$$1 \geq \|\tilde{\boldsymbol{u}}^{(k)}\|_2^2 = 1 - \sum_{i \in I} (u_i^{(k)})^2 \geq 1 - 25\|\boldsymbol{u}^{(k)} - \boldsymbol{p}^*\|_2^2. \tag{53}$$

By the law of cosine, we have

$$\|\boldsymbol{p}^* - \boldsymbol{u}^{(k)}\|_2^2 = \sin^2 \theta(\boldsymbol{u}^{(k)}, \boldsymbol{p}^*) + (1 - \cos\theta(\boldsymbol{u}^{(k)}, \boldsymbol{p}^*))^2 = 2 - 2\cos\theta(\boldsymbol{u}^{(k)}, \boldsymbol{p}^*)$$

$$= 2\left(1 - \sqrt{1 - \sin^2\theta(\boldsymbol{u}^{(k)}, \boldsymbol{p}^*)}\right) = 2\frac{\sin^2\theta(\boldsymbol{u}^{(k)}, \boldsymbol{p}^*)}{1 + \sqrt{1 - \sin^2\theta(\boldsymbol{u}^{(k)}, \boldsymbol{p}^*)}} \tag{54}$$

$$\leq 2\sin^2\theta(\boldsymbol{u}^{(k)}, \boldsymbol{p}^*).$$

Combining equation 53 and equation 54 proves equation 33.

We next prove equation 34,

$$\sin\theta(\tilde{\boldsymbol{u}}^{(k)}, \boldsymbol{p}^*) \leq 6\sqrt{2}\sin\theta(\boldsymbol{u}^{(k)}, \boldsymbol{p}^*).$$

First, note that $\|\tilde{\boldsymbol{u}}^{(k)} - \boldsymbol{u}^{(k)}\|_2^2 = \sum_{i \in I} \left(u_i^{(k)}\right)^2$. We have

$$\sin\theta(\tilde{\boldsymbol{u}}^{(k)}, \boldsymbol{p}^*) \leq \|\tilde{\boldsymbol{u}}^{(k)} - \boldsymbol{p}\|_2 \leq \|\tilde{\boldsymbol{u}}^{(k)} - \boldsymbol{u}^{(k)}\|_2 + \|\boldsymbol{u}^{(k)} - \boldsymbol{p}^*\|_2 \leq 6\|\boldsymbol{u}^{(k)} - \boldsymbol{p}^*\|_2 \tag{55}$$

where the last inequality is from equation 52. Combined with equation 54, we get equation 34.

# H PERFORMANCE ANALYSIS OF ALGORITHM 2

## H.1 PROOF OF LEMMA 1

In this lemma, we show that conditioned on $(\hat{g}_j, \hat{h}_j) = (g_j, h_j)$ for all $j \in [m]$, if $s(1 - s_1) = \Omega\left(\frac{1}{\delta_2 m} \log \frac{n}{\epsilon}\right)$, the estimator $\hat{p}_i$ defined in equation 5,

$$\hat{p}_i = \frac{K}{(K-2)}\left(\frac{1}{s(1-s_1)}\left(\frac{1}{m}\sum_{j=1}^m \mathbb{1}(A_{ij}^2 = \hat{g}_j \text{ or } \hat{h}_j)\right) - \frac{2}{K}\right),$$

guarantees $\mathbb{P}\left(\|\boldsymbol{p} - \hat{\boldsymbol{p}}\|_\infty < \delta_2\right) \geq 1 - \epsilon$ for any $\epsilon > 0$.

Given $(\hat{g}_j, \hat{h}_j) = (g_j, h_j)$ for all $j \in [m]$, since $\boldsymbol{A}^2$ is independent of $(\hat{g}_j, \hat{h}_j)$, we have

$$\mathbb{E}\left[\mathbb{1}(A_{ij}^2 = \hat{g}_j \text{ or } \hat{h}_j)\right] = \mathbb{P}(A_{ij}^2 = \hat{g}_j \text{ or } \hat{h}_j) = s(1-s_1)\left(\frac{K-2}{K}p_i + \frac{2}{K}\right),$$

$$\text{var}\left(\mathbb{1}(A_{ij}^2 = \hat{g}_j \text{ or } \hat{h}_j)\right) \leq s(1-s_1). \tag{56}$$

By applying the Bernstein's inequality, we can show that

$$\mathbb{P}\left(\left|\sum_{j=1}^m \left(\mathbb{1}(A_{ij}^2 = \hat{g}_j \text{ or } \hat{h}_j) - s(1-s_1)\left(\frac{K-2}{K}p_i + \frac{2}{K}\right)\right)\right| > \frac{(K-2)ms(1-s_1)\delta_2}{K}\right)$$

$$\leq \exp\left(-\frac{\frac{1}{2}\left(\frac{(K-2)ms(1-s_1)\delta_2}{K}\right)^2}{ms(1-s_1) + \frac{1}{3}\frac{(K-2)ms(1-s_1)\delta_2}{K}}\right) \leq \exp\left(-\Theta\left(ms(1-s_1)\delta_2^2\right)\right). \tag{57}$$

Thus, if the sampling probability satisfies

$$s(1-s_1) = \Omega\left(\frac{1}{m\delta_2^2}\log\frac{1}{\epsilon}\right), \tag{58}$$

then we can guarantee that $\mathbb{P}(|\hat{p}_i - p_i| < \delta_2) \geq 1 - \epsilon$. By taking the union bound over $i \in [n]$, if the sampling probability satisfies

$$s(1-s_1) = \Omega\left(\frac{1}{m\delta_2^2}\log\frac{n}{\epsilon}\right), \tag{59}$$

then we can guarantee that $\mathbb{P}\left(\|\hat{\boldsymbol{p}} - \boldsymbol{p}\|_\infty < \delta_2\right) \geq 1 - \epsilon$.

## H.2 PROOF OF THEOREM 2

To prove this theorem, we use similar proof techniques from Zhang et al. (2014). Since the work in Zhang et al. (2014) focuses on the recovery of only the ground-truth label for each task, we generalize the techniques to recover not only the ground-truth label but also the most confusing answer.

We first introduce some notations. Let $\mu_{(a,b),k}^{(i,j)}$ denote the probability that a worker $i \in [n]$ gives label $k \in [K]$ for the assigned task $j \in [m]$ of which the top-two answers are $(g_j, h_j) = (a, b)$. Let $\boldsymbol{\mu}_{(a,b)}^{(i,j)} = [\mu_{(a,b),1}^{(i,j)} \quad \mu_{(a,b),2}^{(i,j)} \quad \cdots \quad \mu_{(a,b),K}^{(i,j)}]^\top$. We introduce a quantity that measures the average ability of workers in distinguishing the ground-truth pair of top-two answers $(g_j, h_j)$ from any other pair $(a, b) \in [K]^2 / \{(g_j, h_j)\}$ for the task $j \in [m]$. We define

$$\overline{D}^{(j)} := \min_{(g_j, h_j) \neq (a, b)} \frac{1}{n} \sum_{i=1}^{n} \mathbb{D}_{\mathsf{KL}} \left( \boldsymbol{\mu}_{(g_j, h_j)}^{(i,j)}, \boldsymbol{\mu}_{(a,b)}^{(i,j)} \right); \quad \overline{D} := \min_{j \in [m]} \overline{D}^{(j)}, \tag{60}$$

where $\mathbb{D}_{\mathsf{KL}}(P, Q) := \sum_i P(i) \log(P(i)/Q(i))$ is the KL-divergence between $P$ and $Q$. Note that $\overline{D}^{(j)}$ is strictly positive if $q_j \in (1/2, 1)$ and there exists at least one worker $i$ with $p_i > 0$ for the distribution equation 1, so that $(g_j, h_j)$ can be distinguished from any other $(a, b) \in [K]^2 / \{(g_j, h_j)\}$ statistically. We define $\overline{D}$ as the minimum of $\overline{D}^{(j)}$ over $j \in [m]$, indicating the average ability of workers in distinguishing $(g_j, h_j)$ from any other $(a, b)$ for the most difficult task in the set.

Let us define an event that will be shown holding with high probability,

$$\mathcal{E} : \sum_{i=1}^{n} \sum_{k=1}^{K} \mathbb{1}(A_{ij} = k) \log \left( \frac{\mu_{(g_j, h_j), k}^{(i,j)}}{\mu_{(a,b), k}^{(i,j)}} \right) \geq ns\overline{D}/2 \text{ for all } j \in [m] \text{ and } (a, b) \in [K] \times [K] \backslash (g_j, h_j). \tag{61}$$

Define

$$l_i := \sum_{k=1}^{K} \mathbb{1}(A_{ij} = k) \log \left( \mu_{(g_j, h_j), k}^{(i,j)} / \mu_{(a,b), k}^{(i,j)} \right). \tag{62}$$

We can see that $l_1, \ldots, l_n$ are mutually independent on any value of $(g_j, h_j)$, and each $l_i$ belongs to the interval $[0, \log(1/\rho)]$ where $\mu_{(g_j, h_j), c}^{(i,j)} \geq \rho$ for all $(i, j, g_j, h_j, c) \in [n] \times [m] \times [K]^3$. We can easily show that

$$\mathbb{E} \left[ \sum_{i=1}^{n} l_i \middle| (g_j, h_j) \right] = \sum_{i=1}^{n} s\mathbb{D}_{\mathsf{KL}} \left( \boldsymbol{\mu}_{(g_j, h_j)}^{(i,j)}, \boldsymbol{\mu}_{(a,b)}^{(i,j)} \right). \tag{63}$$

We define

$$D := \sum_{i=1}^{n} \mathbb{D}_{\mathsf{KL}} \left( \boldsymbol{\mu}_{(g_j, h_j)}^{(i,j)}, \boldsymbol{\mu}_{(a,b)}^{(i,j)} \right). \tag{64}$$

The following lemma shows that the second moment of $l_i$ is bounded above by the KL-divergence between the label distribution under $(g_j, h_j)$ pair and the label distribution under $(a, b)$ pair.

**Lemma 6.** *Conditioning on any value of $(g_j, h_j)$, we have*

$$\mathbb{E} \left[ l_i^2 | (g_j, h_j) \right] \leq \frac{2 \log(1/\rho)}{1 - \rho} s\mathbb{D}_{\mathsf{KL}} \left( \boldsymbol{\mu}_{(g_j, h_j)}^{(i,j)}, \boldsymbol{\mu}_{(a,b)}^{(i,j)} \right). \tag{65}$$

The proof of this lemma can be obtained by following the proof of the similar result, Lemma 4 of Zhang et al. (2014).

According to Lemma 6, the aggregated second moment of $l_i$ is bounded by

$$\mathbb{E} \left[ \sum_{i=1}^{n} l_i^2 \middle| (g_j, h_j) \right] \leq \frac{2 \log(1/\rho)}{1 - \rho} \sum_{i=1}^{n} s\mathbb{D}_{\mathsf{KL}} \left( \boldsymbol{\mu}_{(g_j, h_j)}^{(i,j)}, \boldsymbol{\mu}_{(a,b)}^{(i,j)} \right)$$
$$= \frac{2 \log(1/\rho)}{1 - \rho} sD. \tag{66}$$

Thus, applying the Bernstein's inequality, we have

$$\mathbb{P}\left[\sum_{i=1}^{n} l_i \geq sD/2 \middle| (g_j, h_j)\right] \geq 1 - \exp\left(-\frac{\frac{1}{2}(sD/2)^2}{\frac{2\log(1/\rho)}{1-\rho}sD + \frac{1}{3}(2\log(1/\rho))(sD/2)}\right). \tag{67}$$

Since $\rho \leq 1/2$ and $D \geq n\overline{D}^{(j)} \geq n\overline{D}$, combining the above inequality with union bound over $j \in [m]$, we have

$$\mathbb{P}[\mathcal{E}] \geq 1 - mK^2 \exp\left(-\frac{ns\overline{D}}{33\log(1/\rho)}\right). \tag{68}$$

The maximum likelihood estimator finds a pair of $(a, b) \in [K]^2$, $a \neq b$, maximizing

$$(\hat{g}_j, \hat{h}_j) = \operatorname*{arg\,max}_{(a,b)\in[K]^2, a\neq b} \prod_{i=1}^{n} \mathbb{P}(A_{ij}|\boldsymbol{p}, q_j, (a, b))$$

$$= \operatorname*{arg\,max}_{(a,b)\in[K]^2, a\neq b} \sum_{i=1}^{n} \log \mathbb{P}(A_{ij}|\boldsymbol{p}, q_j, (a, b))$$

$$= \operatorname*{arg\,max}_{(a,b)\in[K]^2, a\neq b} \sum_{i=1}^{n} \sum_{k=1}^{K} \mathbb{1}(A_{ij} = k) \log \mu_{(a,b),k}^{(i,j)}. \tag{69}$$

The plug-in MLE in equation 6, on the other hand, finds a pair of $(a, b) \in [K]^2$, $a \neq b$, maximizing

$$(\hat{g}_j, \hat{h}_j) = \operatorname*{arg\,max}_{(a,b)\in[K]^2, a\neq b} \sum_{i=1}^{n} \sum_{k=1}^{K} \mathbb{1}(A_{ij} = k) \log \hat{\mu}_{(a,b),k}^{(i,j)} \tag{70}$$

where $\hat{\mu}_{(a,b),k}^{(i,j)}$ is the estimated probability that a worker $i \in [n]$ gives label $k \in [K]$ for the assigned task $j \in [m]$ of which the top two answers are $(g_j, h_j) = (a, b)$ assuming $p_i = \hat{p}_i$ from equation 5 and $q_j = \hat{q}_j$ from equation 4 in the distribution equation 1. Thus, for the plug-in MLE to correctly find the ground-truth top two answers $(g_j, h_j)$, we need to satisfy the following event:

$$\sum_{i=1}^{n} \sum_{k=1}^{K} \mathbb{1}(A_{ij} = k) \log\left(\hat{\mu}_{(g_j,h_j),k}^{(i,j)}/\hat{\mu}_{(a,b),k}^{(i,j)}\right) \geq 0 \text{ for all } (a, b) \in [K] \times [K] \backslash (g_j, h_j). \tag{71}$$

For any arbitrary $(a, b) \neq (g_j, h_j)$, consider the quantity

$$Q_{(a,b)} := \sum_{i=1}^{n} \sum_{k=1}^{K} \mathbb{1}(A_{ij} = k) \log\left(\hat{\mu}_{(g_j,h_j),k}^{(i,j)}/\hat{\mu}_{(a,b),k}^{(i,j)}\right), \tag{72}$$

which can be written as

$$Q_{(a,b)} = \sum_{i=1}^{n} \sum_{k=1}^{K} \mathbb{1}(A_{ij} = k) \log \frac{\mu_{(g_j,h_j),k}^{(i,j)}}{\mu_{(a,b),k}^{(i,j)}} + \sum_{i=1}^{n} \sum_{k=1}^{K} \mathbb{1}(A_{ij} = k) \left[\log\left(\frac{\hat{\mu}_{(g_j,h_j),k}^{(i,j)}}{\mu_{(g_j,h_j),k}^{(i,j)}}\right) - \log\left(\frac{\hat{\mu}_{(a,b),k}^{(i,j)}}{\mu_{(a,b),k}^{(i,j)}}\right)\right]. \tag{73}$$

Assuming that there exist $\rho > \delta_3$ such that

$$\mu_{(a,b),k}^{(i,j)} \geq \rho \text{ and } |\hat{\mu}_{(a,b),k}^{(i,j)} - \mu_{(a,b),k}^{(i,j)}| \leq \delta_3 \text{ for all } i \in [n], j \in [m], (a, b) \in [K]^2, \tag{74}$$

we have

$$\max_{i\in[n], k\in[K]} \left[\log\left(\frac{\hat{\mu}_{(g_j,h_j),k}^{(i,j)}}{\mu_{(g_j,h_j),k}^{(i,j)}}\right) - \log\left(\frac{\hat{\mu}_{(a,b),k}^{(i,j)}}{\mu_{(a,b),k}^{(i,j)}}\right)\right] \leq 2\log\left(\frac{\rho}{\rho - \delta_3}\right). \tag{75}$$

By the Bernstein's inequality, we also have

$$\mathbb{P}\left[\left|\sum_{i=1}^{n} \sum_{k=1}^{K} \mathbb{1}(A_{ij} = k) - ns\right| > ns/2\right] \leq \exp\left(-\frac{\frac{1}{2}(ns/2)^2}{ns + \frac{1}{3}(ns/2)}\right) = \exp\left(-\frac{3ns}{28}\right). \tag{76}$$

By taking the union bound over $j \in [m]$, we have

$$\mathbb{P}\left[\left|\sum_{i=1}^{n}\sum_{k=1}^{K}\mathbb{1}(A_{ij}=k)-ns\right| > ns/2 \text{ for any } j \in [m]\right] \leq m\exp\left(-\frac{3ns}{28}\right). \qquad (77)$$

Under the intersection of the event $\left|\sum_{i=1}^{n}\sum_{k=1}^{K}\mathbb{1}(A_{ij}=k)-ns\right| \leq ns/2$ for all $j \in [m]$ and the event $\mathcal{E}$, we can guarantee

$$Q_{(a,b)} = \sum_{i=1}^{n}\sum_{k=1}^{K}\mathbb{1}(A_{ij}=k)\log\frac{\mu_{(g_j,h_j),k}^{(i,j)}}{\mu_{(a,b),k}^{(i,j)}} + \sum_{i=1}^{n}\sum_{k=1}^{K}\mathbb{1}(A_{ij}=k)\left[\log\left(\frac{\hat{\mu}_{(g_j,h_j),k}^{(i,j)}}{\mu_{(g_j,h_j),k}^{(i,j)}}\right) - \log\left(\frac{\hat{\mu}_{(a,b),k}^{(i,j)}}{\mu_{(a,b),k}^{(i,j)}}\right)\right]$$
$$\geq \frac{ns\overline{D}}{2} - 3ns\log\left(\frac{\rho}{\rho-\delta_3}\right) \geq ns\left(\frac{\overline{D}}{2} - \frac{3\delta_3}{\rho-\delta_3}\right) > 0 \qquad (78)$$

for every $j \in [m]$ where the last inequality holds if

$$\delta_3 < \rho\frac{\overline{D}}{6+\overline{D}}. \qquad (79)$$

In summary, under that the event $\left|\sum_{i=1}^{n}\sum_{k=1}^{K}\mathbb{1}(A_{ij}=k)-ns\right| \leq ns/2$ for all $j \in [m]$ and the event $\mathcal{E}$ hold, if we have $\delta_3$ such that

$$|\hat{\mu}_{(a,b),k}^{(i,j)} - \mu_{(a,b),k}^{(i,j)}| \leq \delta_3 \text{ for all } i \in [n], j \in [m], (a,b) \in [K]^2 \qquad (80)$$

and

$$\delta_3 < \rho \quad \text{and} \quad \delta_3 < \rho\frac{\overline{D}}{6+\overline{D}}, \qquad (81)$$

then we can guarantee that the plug-in MLE in equation 70 successfully recovers the pair of top two $(g_j, h_j)$ for all the tasks $j \in [m]$. To make the right-hand side of equation 68 and equation 77 less than $\epsilon/2$, it is sufficient to have

$$s = \Omega\left(\frac{\log(1/\rho)\log(mK^2/\epsilon) + \overline{D}\log(m/\epsilon)}{n\overline{D}}\right). \qquad (82)$$

Lastly, when we have

$$\max\{\|\boldsymbol{p}-\hat{\boldsymbol{p}}\|_{\infty}, \|\boldsymbol{q}-\hat{\boldsymbol{q}}\|_{\infty}\} \leq \delta, \qquad (83)$$

we can guarantee that

$$|\hat{\mu}_{(a,b),k}^{(i,j)} - \mu_{(a,b),k}^{(i,j)}| \leq 4\delta := \delta_3. \qquad (84)$$

Thus, it is sufficient to guarantee equation 83 with

$$\delta < \min\left\{\frac{\rho}{4}, \frac{\rho\overline{D}}{4(6+\overline{D})}\right\}. \qquad (85)$$

# I  PROOF OF THEOREM 3

## I.1  PROOF OF PART (A)

To prove this minimax bound, we use the similar arguments from Karger et al. (2014). In particular, we consider a spammer-hammer model such that

$$p_i = \begin{cases} 0, & \text{for } 1 \leq i \leq \lfloor(1-\overline{p})n\rfloor \\ 1, & \text{otherwise.} \end{cases} \qquad (86)$$

Assume that total $l_j$ workers randomly sampled from $[n]$ provide answers for the task $j$. Under the spammer-hammer model, the oracle estimator makes a mistake on task $j$ with probability $(K-1)/K$ if it is only assigned to spammers. When $l_j$ is the number of assignments, we have

$$\mathbb{P}(\hat{g}_j \neq g_j) = \frac{K-1}{K}(1-\overline{p})^{l_j}. \qquad (87)$$

By convexity and using Jensen's inequality, the average probability of error is lower bounded by

$$\frac{1}{m} \sum_{j \in [m]} \mathbb{P}(\hat{g}_j \neq g_j) \geq \frac{K-1}{K}(1-\overline{p})^l \tag{88}$$

where $\frac{1}{m} \sum_{i \in [m]} l_i \leq l$. By assuming $\overline{p} \leq 2/3$, we have $(1-\overline{p}) \geq e^{-(\overline{p}+\overline{p}^2)}$. Thus,

$$\min_{\hat{g}} \max_{p \in \mathcal{F}_{\overline{p}}, \, g \in [K]^m} \frac{1}{m} \sum_{j \in [m]} \mathbb{P}(\hat{g}_j \neq g_j) \geq \frac{K-1}{K} e^{-(\overline{p}+\overline{p}^2)l} \geq \frac{K-1}{K} e^{-2\overline{p}l}. \tag{89}$$

The inequality in equation 89 implies that if $l$ is less than $\frac{1}{2\overline{p}} \log\left(\frac{K-1}{K\epsilon}\right)$, then no algorithm can make the minimax error in equation 89 less than $\epsilon$. Since the average number of queries per task in our model is $ns$, it implies that it is necessary to have $s = \Omega\left(\frac{1}{\|p\|_2^2} \log \frac{1}{\epsilon}\right)$.

## I.2 PROOF OF PART (B)

To prove the second part of the theorem, we use proof techniques from Zhang et al. (2014), but generalizes the results for pair of top two answers. We assume that $j_c \in [m]$, $(g_c, h_c) \in [K]^2$ and $(a_c, b_c) \in [K]^2$ are the task index and the pairs of labels such that

$$\overline{D} = \frac{1}{n} \sum_{i=1}^n \mathbb{D}_{\mathsf{KL}}\left(\boldsymbol{\mu}_{(g_c,h_c)}^{(i,j_c)}, \boldsymbol{\mu}_{(a_c,b_c)}^{(i,j_c)}\right) \tag{90}$$

for $\overline{D}$ defined in equation 60.

Let $\mathbb{Q}$ be a uniform distribution over the set $\{(g_c, h_c), (a_c, b_c)\}^m$. For any $(\hat{g}, \hat{h})$, we have

$$\max_{\substack{(\boldsymbol{v},\boldsymbol{u}) \in [K]^m \times [K]^m \\ v_j \neq u_j, \forall j[m]}} \mathbb{E}\left[\sum_{j=1}^m \mathbb{1}((\hat{g}_j, \hat{h}_j) \neq (g_j, h_j)) \Big| (\boldsymbol{g}, \boldsymbol{h}) = (\boldsymbol{v}, \boldsymbol{u})\right]$$

$$\geq \sum_{j=1}^m \sum_{(\boldsymbol{v},\boldsymbol{u}) \in \{(g_c,h_c),(a_c,b_c)\}^m} \mathbb{Q}((\boldsymbol{v},\boldsymbol{u}))\mathbb{E}\left[\mathbb{1}((\hat{g}_j, \hat{h}_j) \neq (g_j, h_j)) \Big| (\boldsymbol{g}, \boldsymbol{h}) = (\boldsymbol{v}, \boldsymbol{u})\right] \tag{91}$$

Let $\boldsymbol{A} := \{A_{ij} : i \in [n], j \in [m]\}$ be the set of observations. Define two probability measures $\mathbb{P}_0$ and $\mathbb{P}_1$, such that $\mathbb{P}_0$ is the measure of $\boldsymbol{A}$ conditioned on $(g_j, h_j) = (g_c, h_c)$, while $\mathbb{P}_1$ is that on $(g_j, h_j) = (a_c, b_c)$. Then, we can have

$$\sum_{(\boldsymbol{v},\boldsymbol{u}) \in \{(g_c,h_c),(a_c,b_c)\}^m} \mathbb{Q}((\boldsymbol{v},\boldsymbol{u}))\mathbb{E}\left[\mathbb{1}((\hat{g}_j, \hat{h}_j) \neq (g_j, h_j)) \Big| (\boldsymbol{g}, \boldsymbol{h}) = (\boldsymbol{v}, \boldsymbol{u})\right]$$

$$= \mathbb{Q}((g_j, h_j) = (g_c, h_c))\mathbb{P}_0((\hat{g}_j, \hat{h}_j) \neq (g_c, h_c)) + \mathbb{Q}((g_j, h_j) = (a_c, b_c))\mathbb{P}_1((\hat{g}_j, \hat{h}_j) \neq (a_c, b_c))$$

$$\geq \frac{1}{2} - \frac{1}{2}\|\mathbb{P}_0 - \mathbb{P}_1\|_{\mathsf{TV}}$$

$$\geq \frac{1}{2} - \frac{1}{4}\sqrt{\mathbb{D}_{\mathsf{KL}}(\mathbb{P}_0, \mathbb{P}_1)}. \tag{92}$$

where the second to the last inequality is by Le Cam's method and the last inequality is by Pinsker's inequality.[4]

Conditioned on $(g_j, h_j)$, the set of random variables $A_j := \{A_{ij} : i \in [n]\}$ are independent of $\boldsymbol{A} \backslash A_j$ for both $\mathbb{P}_0$ and $\mathbb{P}_1$, and thus

$$\mathbb{D}_{\mathsf{KL}}(\mathbb{P}_0, \mathbb{P}_1) = \mathbb{D}_{\mathsf{KL}}(\mathbb{P}_0(A_j), \mathbb{P}_1(A_j)) + \mathbb{D}_{\mathsf{KL}}(\mathbb{P}_0(\boldsymbol{A} \backslash A_j), \mathbb{P}_1(\boldsymbol{A} \backslash A_j)) = \mathbb{D}_{\mathsf{KL}}(\mathbb{P}_0(A_j), \mathbb{P}_1(A_j)) \tag{93}$$

---

[4]The total variation distance between probability distributions $P$ and $Q$ defined on a set $\mathcal{X}$ is defined as the maximum difference between probabilities they assign on subsets of $\mathcal{X}$: $\|P - Q\|_{\mathsf{TV}} := \sup_{\mathcal{A} \subset \mathcal{X}} |P(\mathcal{A}) - Q(\mathcal{A})|$.

where $\mathbb{P}(X)$ denote the distribution of $X$ with respect to the probability measure $\mathbb{P}$. Given $(g_j, h_j)$, since $A_{1j}, \ldots, A_{nj}$ are independent, we can show that

$$
\begin{aligned}
\mathbb{D}_{\mathsf{KL}}(\mathbb{P}_0(A_j), \mathbb{P}_1(A_j)) &= \sum_{i=1}^{n} \mathbb{D}_{\mathsf{KL}}(\mathbb{P}_0(A_{ij}), \mathbb{P}_1(A_{ij})) \\
&= \sum_{i=1}^{n} \left( (1-s) \log \frac{1-s}{1-s} + s\mathbb{D}_{\mathsf{KL}}\left( \boldsymbol{\mu}_{(g_c,h_c)}^{(i,j)}, \boldsymbol{\mu}_{(a_c,b_c)}^{(i,j)} \right) \right) \\
&\geq sn\overline{D}.
\end{aligned}
\tag{94}
$$

Combining equation 91– equation 94, we have

$$
\begin{aligned}
&\max_{\substack{(\boldsymbol{v},\boldsymbol{u})\in[K]^m\times[K]^m \\ v_j\neq u_j, \forall j[m]}} \mathbb{E}\left[ \frac{1}{m}\sum_{j=1}^{m} \mathbb{1}((\hat{g}_j, \hat{h}_j) \neq (g_j, h_j)) \Big| (\boldsymbol{g}, \boldsymbol{h}) = (\boldsymbol{v}, \boldsymbol{u}) \right] \\
&\geq \frac{1}{2} - \frac{1}{4}\sqrt{sn\overline{D}}.
\end{aligned}
\tag{95}
$$

Thus, if $s \leq \frac{1}{4n\overline{D}}$, then the above inequality is lower bounded by $3/8$. This completes the proof.

## J  USEFUL INEQUALITIES

In this section, we summarize the useful inequalities used in the proof of the main results.

The following inequality, which appeared in Bandeira & Van Handel (2016) provides a non-asymptotic spectral norm bound for random matrices with independent random entries.

**Theorem 4** (Spectral norm bound of a random matrice with independent entries). *Consider a random matrix $\boldsymbol{X} \in \mathbb{R}^{n\times m}$, whose entries are independently generated and obey*

$$
\mathbb{E}[X_{i,j}] = 0, \quad \text{and} \quad |X_{i,j}| \leq B, \quad 1 \leq i \leq n, \ 1 \leq j \leq m.
\tag{96}
$$

*Define*

$$
\nu := \max\left\{ \max_i \sum_j \mathbb{E}[X_{i,j}^2], \ \max_j \sum_i \mathbb{E}[X_{i,j}^2] \right\}.
\tag{97}
$$

*Then there exists some universal constant $c > 0$ such that for any $t > 0$,*

$$
\mathbb{P}\left\{ \|\boldsymbol{X}\| \geq 4\sqrt{\nu} + t \right\} \leq (n+m)\exp\left( -\frac{t^2}{cB^2} \right).
\tag{98}
$$

We also present a useful corollary of Theorem 4, which can be shown from equation 98 by setting $\tilde{c} = \sqrt{9c}$ and $t = B\sqrt{9c\log(n+m)}$.

**Corollary 3** (Corollary of Theorem 4). *If $\mathbb{E}[X_{i,j}^2] \leq \sigma^2$ for all $i, j$ and satisfying conditions in Theorem 4, then we have*

$$
\|\boldsymbol{X}\| \leq 4\sigma\sqrt{\max(m,n)} + \tilde{c}B\sqrt{\log(n+m)}
\tag{99}
$$

*with probability $1 - (n+m)^{-8}$ for some constant $\tilde{c} > 0$.*

We next summarize the eigenspace perturbation theory for asymmetric matrices with singular value composition (SVD). Suppose $\boldsymbol{X} := [\boldsymbol{X}_0, \boldsymbol{X}_1]$ and $\boldsymbol{Z} := [\boldsymbol{Z}_0, \boldsymbol{Z}_1]$ are orthonormal matrices. When we define the distance between two subspaces $\boldsymbol{X}_0$ and $\boldsymbol{Z}_0$ by

$$
\mathsf{dist}(\boldsymbol{X}_0, \boldsymbol{Z}_0) := \|\boldsymbol{X}_0\boldsymbol{X}_0^\top - \boldsymbol{Z}_0\boldsymbol{Z}_0^\top\|,
\tag{100}
$$

then we have

$$
\mathsf{dist}(\boldsymbol{X}_0, \boldsymbol{Z}_0) = \|\boldsymbol{X}_0^\top \boldsymbol{Z}_1\| = \|\boldsymbol{Z}_0^\top \boldsymbol{X}_1\|.
\tag{101}
$$

Given $\|\boldsymbol{X}_0^\top \boldsymbol{Z}_0\| \leq 1$, we write SVD of $\boldsymbol{X}_0^\top \boldsymbol{Z}_0 \in \mathbb{R}^{r \times r}$ as $\boldsymbol{X}_0^\top \boldsymbol{Z}_0 := \boldsymbol{U} \cos \Theta \boldsymbol{V}^\top$ where $\cos \Theta = \mathrm{diag}(\cos \theta_1, \ldots, \cos \theta_r)$. We call $\{\theta_1, \ldots, \theta_r\}$ principal angles between $\boldsymbol{X}_0$ and $\boldsymbol{Z}_0$. Then, we have

$$\|\boldsymbol{X}_0^\top \boldsymbol{Z}_1\| = \|\sin \Theta\| = \max\{|\sin \theta_1|, \cdots, |\sin \theta_r|\}. \tag{102}$$

Let $\boldsymbol{M}^*$ and $\boldsymbol{M} = \boldsymbol{M}^* + \boldsymbol{E}$ be two matrices in $\mathbb{R}^{n \times m}$ with $n \leq m$, whose SVD are represented by $\boldsymbol{M}^* = \sum_{i=1}^n \sigma_i^* \boldsymbol{u}_i^* \boldsymbol{v}_i^{*\top}$ and $\boldsymbol{M} = \sum_{i=1}^n \sigma_i \boldsymbol{u}_i \boldsymbol{v}_i^\top$, where $\sigma_1 \geq \cdots \geq \sigma_n$ (resp. $\sigma_1^* \geq \cdots \geq \sigma_n^*$). Let us define

$$\boldsymbol{U}_0 := [\boldsymbol{u}_1, \cdots, \boldsymbol{u}_r] \in \mathbb{R}^{n \times r}, \quad \boldsymbol{V}_0 := [\boldsymbol{v}_1, \cdots, \boldsymbol{v}_r] \in \mathbb{R}^{m \times r}. \tag{103}$$

The matrices $\boldsymbol{U}_0^*$ and $\boldsymbol{V}_0^*$ are defined analogously.

**Theorem 5** (Wedin $\sin \Theta$ Theorem). *If $\|\boldsymbol{E}\| < \sigma_r^* - \sigma_{r+1}^*$, then one has*

$$\max\{\|\mathrm{dist}(\boldsymbol{U}_0, \boldsymbol{U}_0^*)\|, \|\mathrm{dist}(\boldsymbol{V}_0, \boldsymbol{V}_0^*)\|\} \leq \frac{\sqrt{2}\|\boldsymbol{E}\|}{\sigma_r^* - \sigma_{r+1}^* - \|\boldsymbol{E}\|}, \tag{104}$$

*where $\boldsymbol{U}_0^*$ ($\boldsymbol{V}_0^*$) and $\boldsymbol{U}_0$ ($\boldsymbol{V}_0$) are subspaces spanned by the largest $r$ left (right) singular vectors of $\boldsymbol{M}^*$ and $\boldsymbol{M}$, respecively.*

Lastly, we also write down two useful concentration inequalities.

**Theorem 6** (Hoeffding). *Let $X_1, X_2, \ldots, X_n$ be independent random variables such that $X_i \in [a_i, b_i]$ for $1 \leq i \leq n$. Then, we have*

$$\mathbb{P}\left[\left|\sum_{i=1}^n (X_i - \mathbb{E}[X_i])\right| > t\right] \leq 2 \exp\left(-\frac{2t^2}{\sum_{i=1}^n (b_i - a_i)^2}\right). \tag{105}$$

**Theorem 7** (Bernstein). *Let $X_1, X_2, \ldots, X_n$ be independent random variables such that $X_i \in [a_i, b_i]$ for $1 \leq i \leq n$. Let $C := \max_{1 \leq i \leq n}(b_i - a_i)$ and $\sigma^2 = \sum_{i=1}^n \mathrm{var}(X_i)$. Then we have*

$$\mathbb{P}\left[\left|\sum_{i=1}^n (X_i - \mathbb{E}[X_i])\right| > t\right] \leq 2 \exp\left(-\frac{t^2/2}{\sigma^2 + C \cdot t/3}\right). \tag{106}$$

