# OpenReview forum: "Recovering Top-Two Answers and Confusion Probability in Multi-Choice Crowdsourcing"
_ICLR.cc/2023/Conference — Submitted to ICLR 2023_

### Official Review · Reviewer_UdYX · 2022-10-23

**Confidence:** 4
**Correctness:** 3
**Technical Novelty And Significance:** 3
**Empirical Novelty And Significance:** 3
**Recommendation:** 6

**Clarity, Quality, Novelty And Reproducibility:**

Clarity,
In this paper, the introduction of the model and algorithm is clear, but the logic is not strong in introducing the significance and background of the work in this paper. There are some problems as fellow：
1. For example, the abstract part lacks a clear description of the background and significance of the research.
2. I have a question, when the confusion probability is not 0.5, how to understand the confusion probability and how its value affects the result?

Quality,
The theoretical and experimental parts of this paper are relatively complete, but there are some detailed problems as follows:
1. The organization of related work in the introduction is strange.
2. The latest works are barely presented in the relevant work section
3. In the conclusion section, the reference is very strange

Novelty
This paper proposes a new multi-choice crowdsourcing task model and proposes a two-stage algorithm to recover the top two answers. The method recovers the top-two answers and the confusion probability of each task at the minimax optimal convergence rate. The method in this paper is quite innovative.

Reproducibility
According to the content of this paper, the method in this paper is reproducible

**Strength And Weaknesses:**

Please refer to Summary Of The Review.

**Summary Of The Paper:**

In order to simulate the effect of confusion in multi-choice crowdsourcing problems, this paper proposes a new multi-choice crowdsourcing task model and provides a two-stage inference algorithm to recover the first two answers and the confusion probability of each task. Finally, it shows the potential application of the proposed algorithm.

**Summary Of The Review:**

Most previous multiple-choice crowdsourcing models cannot simulate this kind of confusion of plausibility. This paper provides an effective two-stage inference algorithm to reply to the first two plausible answers and confusion probability and can achieve the optimal convergence speed. This method is compared with other recent crowdsourcing algorithms and achieves superior results. This paper is relatively innovative.

---

> ### Author Response · Authors · 2022-11-15
> **Response to Reviewer UdYX**
>
> We sincerely thank the reviewer for the constructive feedback. We tried our best to address the reviewer’s concerns and questions. Hope this response answers the reviewer’s questions.
>
> ---------
> **[1. The abstract part lacks a clear description of the background and significance of the research.]**
>
> In the revised version, we updated the abstract to better describe the importance of the crowdsourcing problem and the main contribution of this work.
>
> ---------
> **[2. When the confusion probability is not 0.5, how to understand the confusion probability and how its value affects the result?]**
>
> The confusion probability $q_j\in[0.5,1]$ indicates how plausible the confusable answer is compared to the ground-truth answer in each task $j$. In our model, this value can be different over the tasks. Even for a perfectly reliable worker with $p_i=1$, if a task has the confusion probability $q_j<1$, the chance that the answer is equal to the ground truth/confusable class is equal to $q_j$ and $1-q_j$, respectively. Thus, when the confusion probability is equal to 0.5 for a certain task, a worker will give the ground-truth answer and the confusable answer with equal probabilities. On the other hand, as $q_j$ increases, the level of confusion between the top-two most plausible answers decreases, and for a task with $q_j=1$, a perfectly reliable worker may provide only the ground-truth answer with probability 1.
>
> By modeling and estimating the chance of confusion for each task, we decompose the erroneous behavior of workers into two factors, low reliability of a worker and confusion originated from task difficulty. This modeling allows us to better estimate the worker reliabilities, and it enables our inference algorithms to achieve better performances (smaller error probabilities) at a given number of queries per task, as demonstrated by the theorems and empirical results.
>
> ---------
> **[3. Related work: the latest works are barely presented in the relevant work section.]**
>
> In the related work section, we presented the most relevant works to this paper, to highlight the major differences of our work in terms of the modeling and algorithms. We included the most recent works such as [a]-[d] in the manuscript as well.
>
> As the Reviewer mv6B suggested, we also included the recent algorithm developed in [a] as an additional baseline of our experiments, presented in Fig. 1 and 2.
>
> [a] Ma et al., Adversarial Crowdsourcing Through Robust Rank-One Matrix Completion, NeurIPS 2020.
>
> [b] Ibrahim et al. Crowdsourcing via Pairwise Co-occurrences: Identifiability and Algorithms, Neurips 2019.
>
> [c] Tian et al., Max-Margin Majority Voting for Learning from Crowds, NeurIPS 2015.
>
> [d] Li et al., Exploiting Worker Correlation for Label Aggregation in Crowdsourcing, ICML 2019.

---

### Official Review · Reviewer_pX1G · 2022-10-24

**Confidence:** 2
**Correctness:** 4
**Technical Novelty And Significance:** 4
**Empirical Novelty And Significance:** 4
**Recommendation:** 8

**Clarity, Quality, Novelty And Reproducibility:**

The writing was clear and the experiments are reproducible. The proposed model and related methods seem to be novel.

**Strength And Weaknesses:**

I found that the paper provided a thorough description of the existing methods (although I am not an expert on this), clearly characterized the proposed model, and contained an extensive description of the the theoretical and empirical properties of the algorithm. The experiments seemed to be reasonable. I only have the following minor comments:

* 10 repetitions of the experiments to estimate the standard deviation is not enough to get accurate estimates (neither of the mean nor of its sampling variation). This should be increased. In addition, please report the 95% confidence intervals. Based on the reported errors, the intervals might reveal that the performances of the algorithms are not significantly different. The shaded regions in Figure 2a should be described.
* What happens in the "easiest" case where the q=1? Does any of the other algorithms outperform Toptwo2 or Toptwo1?
* Can the model be easily extended to more than two plausible tasks?
* How does performance vary in settings where crowdworkers are all experts with similar ability (p=1)?
* Details about IRB approval for the experiment that has been run should be provided.

Typos: * "ssume" in footnote 2. "achieveed" in page 8.
* "lower bound for the performance" -> upper bound for the performance and lower bound for the error probably?
* \eta is described in footnote 2 and then appears in algorithm 1. It might be worth reintroducing it.

**Summary Of The Paper:**

The paper proposes a "top-two" model in which crowdsourcing tasks admit two plausible answers (the ground truth and a "confusing" answer). The authors argue and show that this model can be applied to many of the crowdsourcing tasks used for existing public datasets. In this framework, they describe a two-stage algorithm to infer the two types of answers and the workers' abilities. The first stage uses SVD on a transformation of a split of the workers x task answers matrix to estimate the difficulty of the task (defined in their eq 1). The second stage obtains the final estimates of the workers' abilities and task difficulty via maximum likelihood. The authors conduct a performance analysis of the algorithm and show that it recovers the two most plausible answers. Lastly, they conduct experiments on both synthetic and real-world datasets, including training a neural network with the top two answers rather than soft labels or the majority vote.

**Summary Of The Review:**

The authors propose to model the difficulty of crowdsourcing tasks with two plausible answers and carefully details the properties of the proposed algorithm in this framework. I only have minor concerns and questions about the existing work that I hope the authors can address in their answer.

---

> ### Author Response · Authors · 2022-11-15
> **Response to Reviewer pX1G**
>
> We sincerely thank the reviewer for the constructive feedback. We tried our best to address the reviewer’s questions.
>
> ---------
> **[1. Increase the number of repetitions in the experiments to estimate the mean and the standard deviation. In addition, please report the 95% confidence intervals. The shaded regions in Figure 2a should be described.]**
>
> We updated the figures in the manuscript (Fig. 1, 6 and 7) by increasing the number of repetitions in the experiments from 10 to 30. The shaded regions now indicate the 95\% confidence interval.
>
> In Fig. 2a, the shaded regions also indicate the 95\% confidence interval of the color dataset experiments. After collecting the data, we subsampled it to simulate how the prediction error decreases as the number of responses per task increases. To generate each data point in Fig. 2a, we randomly sampled the color datasets 10 times and reported the average performance with 95\% confidence interval. We added this detail in the manuscript.
>
>
> ---------
> **[2. What happens in the "easiest" case where the q=1? Do any of the other algorithms outperform Toptwo2 or Toptwo1?]**
>
> For the easiest case (when all the tasks have $q_j=1$), our model is exactly the same as the single-coin Dawid-Skene model, where the error probability of worker responses only depends on the worker skill but not on the task difficulty. For this case, our algorithm still achieves the minimax optimality in the sample complexity.
>
> More specifically, for a task $j$ with $q_j=1$, it is impossible to recover $h_j$ since $h_j$ cannot be distinguished from the rest of wrong labels $c\in[K]\backslash{\\{g_j\\}}$ statistically from equation (1). For such tasks, we can still guarantee the recovery of $g_j$ with accuracy $P\left(\hat{g}_j=g_j\right)\geq1-\epsilon$ under the conditions in Theorem 1 by Algorithm 1, which is the minimax optimal result by the converse statement in Theorem 3.
>
> ---------
> **[3. Can the model be easily extended to more than two plausible answers?]**
>
> We appreciate the reviewer for the insightful suggestion. The answer is “Yes”. We can generalize both the model and algorithms (Alg. 1 and 2) to estimate the top $T\geq 2$ plausible answers for each task as the following steps.
>
> First, we can generalize the model in (1) by assuming that each task $j\in[m]$ has $T\geq 2$ plausible answers, denoted by $\\{g\_{j1}, g\_{j2}, \cdots, g\_{jT}\\}$. We can assume that $g\_{jt}$ is received with probability $s\left(p\_i q\_{jt}+(1-p\_i)/K\right)$ where $\sum_{t=1}^T q\_{jt}=1$. Then, the rank-1 analysis of the observation matrices reveals that $\Delta r\_j^{(k)}$ in (2) can be represented as below.
> $$
> \\Delta r\_j^{(k)}=\frac{1}{K}-q\_{jt}\text{ for }k=g\_{jt}, t\in[T]
> $$
> $$
> \\Delta r\_j^{(k)}=\frac{1}{K}\text{ otherwise }
> $$
> Remind that Algorithm 1 calculates $\Delta v\_j^{(t)}\approx \\|\textbf{p}\\| r\_j^{(t)}$ for $k\in[K]$, and estimate the top-two most plausible answers by finding the two indices of $k\in[K]$ at which $\Delta v\_j^{(k)}$ has the two smallest values. Similarly, we can estimate top $T$ most plausible answers by finding $T$ indices of $k\in[K]$ at which $\Delta v\_j^{(k)}$ have the $T$ smallest values. Also, we can estimate the confusion probabilities $[q\_{j1}, \dots, q\_{jT}]$ from $\hat{q}\_{jt} = 1/K - \Delta v\_j^{(\hat{g}\_{jt})}/l
> $ where $l$ is the estimate for $\\|\textbf{p}\\|$.
>
> Further, our plug-in MLE estimates in (6) of Algorithm 2 can be extended by
> $$
> 	\text{argmax}\_{ a\_1, a\_2, \cdots, a\_T \in [K]} \sum_{i=1}^n \sum_{t=1}^T \log \left(\frac{K\hat{p}\_i \hat{q}\_{jt}}{1-\hat{p}\_i} + 1\right)  \mathbf{1}(A\_{ij} = a\_t).
> $$
>
> Although theoretical analysis needs to be changed accordingly, the model and algorithms can be easily extended to the general case of $T\geq 2$ plausible answers as above, since the binary-converted observation matrices still enjoy the rank-1 structure. Generalizing the theoretical analysis will be an interesting open problem.
>
> ---------
> **[4. How does performance vary in settings where crowdworkers are all experts with similar ability (p=1)?]**
>
> The ultimate goal of the crowdsourcing is refining the answers obtained from the non-experts. If every worker has $p=1$, the probability distribution of every answer only depends on the task difficulty, and the optimal estimation of the ground-truth answer can be done by the simple majority voting.
>
>
> ---------
> **[5. Details about IRB approval for the experiment that has been run should be provided.]**
>
> We checked that we do not need to apply for an IRB approach for the experiments we conducted at MTurk, since an IRB is required only when the questions are about the interviewee themselves. When we collected the color dataset using Amazon MTurk, we only queried which color is the most similar one to the reference color. Thus, it may not be required to get the IRB approval for the data we collected from MTurk.
>
> ---------
> **[6. Typos]**
>
> We carefully revised the paper to fix all the typos.

---

### Official Review · Reviewer_mv6B · 2022-10-25

**Confidence:** 5
**Clarity, Quality, Novelty And Reproducibility:** Please see detailed comments above.
**Correctness:** 4
**Technical Novelty And Significance:** 2
**Empirical Novelty And Significance:** 2
**Recommendation:** 3

**Strength And Weaknesses:**

This work is technically sound, to show the effectiveness of the proposed method, some theoretical guarantees are provided, also some experiments on both synthetic and real datasets were implemented. However, this work can be further improved in the following aspects:

- The baseline methods used are not state-of-art methods. The most recent crowdsourcing method is from 2018. It would be better to compare with more recent works like [1] and
[1]https://proceedings.neurips.cc/paper/2020/file/f86890095c957e9b949d11d15f0d0cd5-Paper.pdf
[2] https://dl.acm.org/doi/10.5555/3454287.3454992

- All the figures are not vector graphs. It should be revised.

- It is not quite clear how the proposed model and algorithm can be applied to real applications.

- The organization of this work is not clear enough, especially the theoretical analyses part. It would be better to have some discussions after the proposition and theorem.

- I also have concerns about the novelty of the proposed model and algorithm. There already exists some work to divide the prediction in crowdsourcing into two stages, where the first stage is focusing on filtering the top answers, and the second stage is doing the prediction. Also, there exist some works to predict the reliability of the workers to do prediction. The proposed model gives me the impression that it is a combination of these methods. I am not sure how much insight this work can bring to the study of crowdsourcing or the ML community.


**Summary Of The Paper:**

This paper considers the multiple-class crowdsourcing problem, where the confusing answers and confusion probability are considered. To address this problem, a new model was proposed to identify the top-two plausible answers for every task. Based on the proposed model, a new algorithm was also developed. To show the effectiveness of the proposed model, both theoretical guarantees and empirical evaluations are provided.

**Summary Of The Review:**

In my opinion, this work is not ready to be published yet. The baseline methods used are not start-of-art methods. The novelty of this work is kind of limited. Also, the organization of this paper can be further improved.

---

> ### Author Response · Authors · 2022-11-15
> **Response to Reviewer mv6B (3/3)**
>
> **[4. The organization of this work is not clear enough, especially the theoretical analyses part. It would be better to have some discussions after the proposition and theorem.]**
>
> Thank you for the helpful suggestion. We added additional discussions of theoretical results in Appendix E to explain the intuition behind the theoretical statements. Here are some key points.
>
>
> * Theorem 1 asserts that the sample complexity of $\Omega\left(\frac{1}{\delta_1^2 \\|\mathbf{p}\\|_2^2}\log\frac{K}{\epsilon}\right)$ is sufficient to recover the top-two answers $(g_j,h_j)$ for any task $j\in[m]$ and to estimate the confusion probability $q_j$ with accuracy of $|\hat{q}_j-q_j|<\delta_1$ by Algorithm 1 with probability at least $1-\epsilon$. Combined with Theorem 3 part (a), we can see that this sample complexity is the minimax optimal rate for a fixed collective quality of workers, measured by $\\|\mathbf{p}\\|_2^2$.
>
> * It is also worth comparing our algorithm with the simple majority voting (MV) scheme. The MV scheme infers the top-two answers by counting the majority of the received answers. Simple analysis shows that the MV scheme requires the sampling probability $s$ such that $ns= \Theta\left((\frac{1}{n}\sum\_i p\_i)^{-2}\log\frac{1}{\epsilon}\right)$ to recover $(g\_j,h\_j)$ with probability $1-\epsilon$. Remind that Algorithm 1 requires $ns=\Omega\left(\frac{n}{\delta\_1^2 \\|\mathbf{p}\\|\_2^2}\log\frac{K}{\epsilon}\right)$ samples per task. Since $\frac{1}{n}\\|\mathbf{p}\\|_2^2=\frac{1}{n}\sum\_{i} p\_i^2 \geq\left(\frac{1}{n}\sum\_i p\_i\right)^2$ by Cauchy-Schwarz inequality, Algorithm 1 achieves strictly better trade-offs unless $p_i$ is same for all workers $i\in[n]$. As an example, for a spammer-hammer model where $\alpha\in(0,1)$ fraction of workers are hammers with $p_i=1$ and the rest are spammers with $p_i=0$,  Algorithm 1 requires $ns=\Theta\left(\frac{1}{\alpha}\log\frac{1}{\epsilon}\right)$ samples per task, while MV requires $ns=\Theta\left(\frac{1}{\alpha^2}\log\frac{1}{\epsilon}\right)$ samples per task to recover top-two answers with probability $1-\epsilon$.
>
> * Theorem 2 shows that when we have an entrywise bound on the estimated worker reliability vector $\mathbf{p}$ and the task difficulty vector $\mathbf{q}$, the plug-in MLE estimator, used in Algorithm 2, guarantees the recovery of top-two answers if the sampling probability $s=\Omega(\frac{\log(m/\epsilon)}{n\bar{D}})$ where $\bar{D}$, which depends on $(\mathbf{p},\mathbf{q})$, indicates the average reliability of workers in distinguishing the top-two answers from any other pairs for the most difficult task. Combined with Theorem 3 part (b), we can see that this sample complexity is the minimax optimal rate for any $(\mathbf{p},\mathbf{q})$, ignoring the logarithmic terms.
>
> * Combining the conditions for the accurate estimation of model parameters (11) and the convergence of the plug-in MLE (Theorem 2), Corollary 2 shows the condition on the sample complexity to guarantee the performance of Algorithm 2.
>
> ---------
> **[5. Novelty of the proposed model and two-stage algorithm. How much insight can this work bring to the study of crowdsourcing or the ML community?]**
>
>  As the reviewer pointed out, two-stage algorithms are prevalent in statistical inference problems, including the crowdsourcing, where the first stage tries to obtain an initial estimate on the ground-truth labels and the second stage refines the initial estimates by either using the estimates for the unknown model parameters or fixing contradictions on the inference by local refinement. However, our algorithm has the major novelty in that the first stage (Algorithm 1) infers not only the initial estimates for the ground-truth labels but also the 'task difficulty’ for all the tasks. This is enabled by the careful analysis of the right-singular vector $\mathbf{r}^{(k)}$, $k\in[K]$, of the binary-converted observation matrices (Proposition 1),  combined with the `entrywise’ bound for the estimated right singular vector (Lemma 5). We’d like to emphasize that this entrywise bound requires more technicality, compared to general analysis of spectral methods where only the angle between the ground-truth and the estimated singular vectors is bounded.
>
> The main insight this work can bring to the study of crowdsourcing or the ML community is that one can infer not only the ground-truth but also the most confusing answer and the confusion probability of each task in the crowdsourcing tasks, by the proper modeling and inference algorithm, and that the knowledge of the 2nd best answer and the confusion probability brings significant advantages in machine learning applications. As an example, in Section 5.2 and 5.3, we showed that the additional information of the 2nd best and the confusion probability can reveal task difficulty in a meaningful manner and can be used in training neural networks with the soft labels, which results in a better generalization of neural networks.

---

> > ### Comment · Reviewer_mv6B · 2022-11-28
> > **Response**
> >
> > I have read the response of the authors, as well as other reviewers' comments carefully. After consideration, I will continue to keep my current score. I think the current experimental results are still not strong enough. New experiments covered only one dataset, also these new baseline methods still didn't cover the work in 2021 or 2022. Besides, I think the novelty of this work is limited.

---

> ### Author Response · Authors · 2022-11-15
> **Response to Reviewer mv6B (2/3)**
>
> **[3. It is not quite clear how the proposed model and algorithm can be applied to real applications.]**
>
> In the main text, we provided two examples where the proposed model and algorithm can be applied to real applications,  in 1) inferring the difficulty of tasks and 2) using the knowledge of the second best answer and the confusion probability as a useful side information of the datasets in training neural networks. In this response, we also describe 3) how to generalize our algorithm for adaptive crowdsourcing setup.
>
> 1) Inferring the difficulty of tasks
>
> The first example is in inferring the task difficulty from collected human answers. Task difficulty provides useful information in identifying hard vs. easy tasks and in re-examining the validity of the inferred ground-truth answer, e.g., final decision for a paper in the peer-review process, or in designing a problem set with controlled exam difficulty. However, inferring the task difficulty is not an obvious process without a good model for human answers. For example, one might infer that a task is difficult if the correct answer rate of the task is lower than those of other tasks, but it is possible that the task has a lower correct answer rate since the task is assigned to unreliable workers. Thus, decomposing the errors from low reliability of workers and the confusion is important in modeling human answers. Our model in (1) addresses this point and our algorithm (Alg.1) infers the task difficulty with theoretical guarantees (Thm. 1). In Sec. 5.2, by conducting human experiments for the designed color comparison tasks (Fig. 4), we demonstrated that the inferred task difficulty from our algorithm indeed has a correlation with the inherent task difficulty, measured by the color distance metric, as shown in Fig. 2.(b).
>
>
> 2) Supervised learning with top-two soft label
>
> As another useful application of the proposed model, we consider training of deep neural networks with top-two soft labels, where the soft label is designed to include the information of the ground-truth label and the confusable class with confusion probability between the two classes. We trained two deep neural networks, VGG-19 and ResNet18, with the soft-label vectors for images in CIFAR 10H dataset, and compared the performances (test accuracy and loss) with those of the hard-label and full soft-label training.  As shown in Table 2, training with the top-two soft label achieves the higher test accuracy compared to that of hard label training, and achieves competitive performances to (or sometimes even better than) that of the full soft-label training. This result shows the benefit of using the top-two soft labels, the side-information of the 2nd best answer and the confusion probability, in training neural networks.
>
> 3) Adaptive crowdsourcing
>
> The confusion probability of each task can also be used to discriminate the confusable tasks from relatively easy tasks in the online crowdsourcing. During collecting the answers from workers, we can estimate the difficulty of each task and then adaptively assign difficult tasks to more future workers in order to increase the confidence on the inferred answers for the difficult tasks.

---

> ### Author Response · Authors · 2022-11-15
> **Response to Reviewer mv6B (1/3)**
>
> We sincerely thank the reviewer for the constructive feedback. We tried our best to address the reviewer’s concerns and questions. Hope this response answers the reviewer’s questions.
>
> ---------
> **[1. The baseline methods used are not state-of-art methods. It would be better to compare with more recent works.]**
>
> We thank the reviewer for the suggestion. We have updated the manuscript (Fig. 1 and 2)  to include the comparison results to the more recent baseline method, M-MSR (NeurIPS, 2020), as the reviewer suggested. M-MSR algorithm was developed based on the assumption that the majority of workers are governed by the single-coin D&S model while some of the workers are adversarial. When the tasks have heterogeneous difficulty, our algorithm (TopTwo2) still achieves better performance as shown in Fig. 1 and 2.
>
> To compare the performance of our algorithms to other state-of-the-art methods, we additionally conducted a real-dataset experiment to compare our method with four recent baselines, M-MSR [a], MultiSPA [b], Max-Margin MV(M$^3$V) [c], EBCC [d]. MultiSPA [b] finds the confusion matrix and the prior PMF of labels using the second-order statistics based on pairwise co-occurrences. Max-margin MV [c] finds a weight vector for weighted majority voting to guarantee a large margin in aggregated scores between the potential true label and any alternative label. EBCC [d] captures worker correlations by modeling true classes as mixtures of subtypes. We compared the performance of these algorithms with ours for the Web dataset (a public dataset with details in Appendix A) and showed that our algorithm (TopTwo2) achieves the best performance as shown in the table below.
>
> |Method|Prediction error|
> |:----|:----:|
> |TopTwo2 (ours)|0.118|
> |TopTwo1 (ours)|0.233|
> |EBCC|0.231|
> |M$^3$V|0.127|
> |MultiSPA-KL| 0.145|
> |M-MSR|0.309|
>
> Although the four new baselines also achieve competitive performance, these baselines are restricted to the case when every multiple-choice task has a fixed set of choices (classes). On the other hand, our algorithm can be applied to any multiple-choice task with choices varying over tasks, and can recover not only the ground truth but also the 2nd best answer of each task.
>
> [a] Ma et al., Adversarial Crowdsourcing Through Robust Rank-One Matrix Completion, NeurIPS 2020.
>
> [b] Ibrahim et al. Crowdsourcing via Pairwise Co-occurrences: Identifiability and Algorithms, Neurips 2019.
>
> [c] Tian et al., Max-Margin Majority Voting for Learning from Crowds, NeurIPS 2015.
>
> [d] Li et al., Exploiting Worker Correlation for Label Aggregation in Crowdsourcing, ICML 2019.
>
> ---------
> **[2. All the figures are not vector graphs. It should be revised.]**
>
> We appreciate your advice on the format of the figures. We changed all the figures as vector graphs.

---

### Official Review · Reviewer_oTxe · 2022-10-26

**Confidence:** 2
**Correctness:** 4
**Technical Novelty And Significance:** 3
**Empirical Novelty And Significance:** 3
**Recommendation:** 6

**Clarity, Quality, Novelty And Reproducibility:**

The paper is well written. I have seen other approaches to using the confusion of raters versus hard labels.

**Strength And Weaknesses:**

- The authors evaluate on both synthetic and real-world datasets. They also demonstrate how their model can improve neural network performance.
- As I was reading through the paper I wondered why only top-two, the authors provided good explanation in the appendix using analysis of public crowdsourcing datasets.

Weakness
- 5.3, It isn't clear to me that top-two is much better than full distribution.

**Summary Of The Paper:**

In crowd-computing tasks there are often two causes for wrong answers: 1) low reliability of worker (i.e. possibly a spammer or someone making a random guess), 2) confusion due to task (or question) difficulty (i.e. the question is simply harder than others or the potential answers could be confusing). The David-Skene model only captures the worker reliability portion of this. The authors have built a model for multi-choice crowdsourcing, that infers the top-two answers and the confusion probability. This would benefit downstream uses of these crowdsource tasks by providing the most plausible answer other than the ground truth and how plausible that second answer is. The authors show how using this top-two information can add neural network training versus using a single hard label.

**Summary Of The Review:**

Overall it was a well written paper. The evaluations on the synthetic and real-world datasets were well done. I am unsure of the results of the neural network training.

---

> ### Author Response · Authors · 2022-11-15
> **Response to Reviewer oTxe**
>
> We sincerely thank the reviewer for the constructive feedback. We tried our best to address the reviewer’s concerns and questions. Hope this response answers the reviewer’s questions.
>
> ---------
> **[1. Sec. 5.3 Training Neural Network with Soft Labels: It isn't clear to me that the top-two is much better than the full distribution.]**
>
> In Table 2 of Sec. 5.3, we reported the test accuracy/loss of two networks (VGG19 and ResNet18) trained with CIFAR10H dataset for three different labeling schemes, 1) hard label, 2) top-two label and 3) full-lagel distribution. By the experiment, we aimed to demonstrate the advantage of knowing the 2nd most probable label and the confusion probability of each instance in training a neural network. Indeed, we could show that using the top-two label consistently achieves a better generalization capability, i.e. 1.56% and 4.09% higher test accuracy in VGG-19 and ResNet18, compared to those of hard labels.
>
> As the reviewer pointed out, training with the top-two labels does not always perform better than the training with the full-label distribution, e.g. the result for ResNet18. However, we argue two important benefits of using the top-two label.
>
> First, training with the top-two label is much more cost efficient in terms of the sample complexity compared to the full label distribution. In fact, obtaining an accurate estimate of the full label distribution requires much higher sample complexity than recovering only the ground truth or top-two answers. For example, CIFAR10H dataset, where the full label distributions for 10,000 images are provided, was constructed by collecting on average 50 judgements per image on average, which is about 5-10 times higher sample complexity than usual crowdsourcing datasets (Table 4 in Appendix A). Thus, if the top-two label can achieve a performance comparable to that of the full label distribution, using the top-two label will be a much more cost efficient choice.
>
> Second, training with the top-two label is much more robust against noise in the crowd answers, compared to that of the full-label distribution. The dataset used in the experiments of Table 2 is collected from workers whose reliability is close to 95%, so that the full label distribution is in fact almost the same as the top-two distribution and we cannot see the significant differences between the two. To compare the robustness against the label noise, we conducted an additional experiment by adding different portions of random labels to the original CIFAR10 dataset. In the table below, we compare how the test accuracy degrades as the corruption portion increases between the top-two labeling and the full distribution. In the experiment, we added the responses from spammers, who provide random labels on each image, to the original dataset, with the varying ratio [0.1, 0.2, 0.3, 0.4, 0.5]. (e.g, if the ratio of the spammer is 0.5, it means that we added the number of responses from spammers the same as that of the original dataset.)
>
>
> |Corruption ratio|ResNet18(top2)|ResNet18(full)|VGG-19(top2)|VGG-19(full)|
> |:----|:----:|:----:|:----:|:----:|
> |0.1|80.18$\pm$1.30|80.73$\pm$0.79|78.90$\pm$0.72|78.67$\pm$1.45|
> |0.2|80.30$\pm$1.81|79.79$\pm$0.59|79.10$\pm$0.64|78.65$\pm$0.91|
> |0.3|79.80$\pm$0.44|79.23$\pm$0.79|79.08$\pm$1.22|77.80$\pm$1.08|
> |0.4|79.05$\pm$0.78|76.82$\pm$0.75|79.15$\pm$1.46|77.40$\pm$1.09|
> |0.5|78.40$\pm$0.96|75.88$\pm$0.93|78.22$\pm$0.69|76.11$\pm$1.53|
>
> As shown in the table, training with the top-two labels is much more robust against the noise in the crowd answers, compared to that of the training with the full-label distribution. This is because the training with the full-label distribution tries to fit the model to all the collected answers, which can include noisy answers. On the other hand, training with the top-two labels is more robust against the label noise, since it focuses on the simple yet meaningful side information, the ground-truth label and the most confusing label with the ratio between the two in the collected answers.
>
> [a] Peterson et al., Human uncertainty makes classification more robust, ICCV, 2019.

---

### Decision · Program_Chairs · 2023-01-20

**Decision:**

Reject

**Justification For Why Not Higher Score:**

The paper had substantial disagreement from reviewers, although the AC and the Reject reviewer had more confidence in their reviews.  The Accept reviewer was not confident.

**Justification For Why Not Lower Score:**

N/A


**Metareview: Summary, Strengths And Weaknesses:**

Training on the top and a second label is certainly a useful idea, and its simple. Several more complex models have been developed for this task.  Experiments are done on predicting difficulty and on considering the training of top 2 labels versus other scenarios, both worthy applications of the idea.  The authors did a good job of addressing some of the reviewers concerns.

Comparisons misses some crowd-sourcing work published since 2020.   Comparisons also missed prior work and survey papers on related tasks of estimating difficulty, estimating subjectivity, estimating skill levels, which are inter-related and make the seemingly black-and-white task of estimating difficulty more nuanced.  However, some additional work such as Ma etal 2020 was compared against.

The experimental results are good demonstrating the system is working, but due to lack of other comparisons they may not be state of the art.  It is a good simple idea, however, so I expect with this comparison the paper- will be much stronger.

Author gives further results which would be better placed in the main paper since experiments in main paper can be improved.


**Summary Of Ac-Reviewer Meeting:**

The meeting was not completed. The best attempts were conducted through OpenReview and emails to ensure that reviewers, AC and SAC could freely speak out their opinions on the rejection decision, and received no objections.